# The H2A.Z histone variant integrates Wnt signaling in intestinal epithelial homeostasis

Jérémie Rispal[1], Lucie Baron[1], Jean-François Beaulieu [2], Martine Chevillard-Briet[1], Didier Trouche[1,3] & Fabrice Escaffit[1,3]

The Tip60/p400 chromatin-modifying complex, which is involved in the incorporation and post-translational modification of the H2A.Z histone variant, regulates cell proliferation and important signaling pathways, such as Wnt. Here, we study the involvement of H2A.Z in intestinal epithelial homeostasis, which is dependent on the finely-tuned equilibrium between stem cells renewal and differentiation, under the control of such pathway. We use cell models and inducible knock-out mice to study the impact of H2A.Z depletion on intestinal homeostasis. We show that H2A.Z is essential for the proliferation of human cancer and normal intestinal crypt cells and negatively controls the expression of a subset of differentiation markers, in cultured cells and mice. H2A.Z impairs the recruitment of the intestine-specific transcription factor CDX2 to chromatin, is itself a target of the Wnt pathway and thus, acts as an integrator for Wnt signaling in the control of intestinal epithelial cell fate and homeostasis.

[1] LBCMCP, Centre de Biologie Intégrative (CBI), Université de Toulouse, CNRS, UPS, Toulouse 31062, France. [2] Laboratory of Intestinal Physiopathology, Department of Anatomy and Cell Biology, Faculty of Medicine and Health Sciences, Université de Sherbrooke, Sherbrooke J1H 5N4 QC, Canada. [3]These authors jointly supervised this work: Didier Trouche, Fabrice Escaffit. Correspondence and requests for materials should be addressed to F.E. (email: fabrice.escaffit@univ-tlse3.fr)

Epigenetics and chromatin-modifying enzymes are known to be major regulators of physio-pathological processes. We recently showed[1] that the Tip60/p400 enzymatic complex plays a critical role in colon cancer, regulating susceptibility to chemically-induced pre-neoplastic lesions and adenomas. We demonstrated that these effects are dependent on the Wnt pathway, a well-known signaling pathway involved in cancer, but also critical for the development and homeostasis of the normal intestine.

The Tip60/p400 complex is a multimolecular complex containing several chromatin-modifying enzymes[2], such as the Tip60 histone acetyltransferase or the p400 ATPase, which participates in the incorporation of the histone variant H2A.Z[3,4]. This protein is one of two histone variants, which are conserved from yeast to human and its enrichment at the Transcription Start Site (TSS) of genes is dependent on their expression levels, indicative of a major role in gene transcription. Recent studies suggest or point out the role of H2A.Z incorporation dynamics in several biological processes, such as senescence, stem cell specification or the learning process[5–7].

The enzymatic activities present in the Tip60 complex participate in the control of transcription in a complex manner. Indeed, while histone acetylation by Tip60 is globally associated with the activation of transcription[8], the incorporation of H2A.Z can differently affect gene expression depending on the gene, chromatin context or post-translational modification of H2A.Z itself[3,5,9–11].

p400 and H2A.Z have been shown to be important for cell proliferation, since their depletion leads to proliferation arrest in many cell types[1,3,10–14]. We previously demonstrated that p400 favours some pro-proliferative pathways: in particular, p400 plays a central role in the control of Wnt activity by preventing oxidative stress[15], which antagonizes the activity of the Wnt pathway[16], as well as through the direct regulation of Wnt target genes.

In mammals, intestinal stem cell niche maintenance and terminal differentiation of epithelial lineages are dependent on the Wnt pathway, in cooperation with other signaling pathways, such as BMP[17] for example. Intestinal stem cells are located in the crypts, interspersed with Paneth cells, which serves as a niche. These stem cells lead to the production of progenitor cells, which actively proliferate. Migration of these cells into the upper compartments of the intestinal epithelium leads to their differentiation into specific intestinal lineages, including enterocytes, goblet cells, or entero-endocrine cells. This process is controlled by intestine-specific transcription factors, such as members of the CDX homeobox gene family, which are crucial for embryogenesis and differentiation of the intestinal epithelium[18,19]. Indeed, CDX1 and CDX2 are essential for the maintenance of intestinal identity and regulate compartment-specific differentiation[20]. For instance, CDX2 functionally interacts with other transcription factors, such as HNF1α and GATA4, to regulate the expression of terminal differentiation markers of enterocytes (Sucrase-Isomaltase (SI), Lactase-Phlorizin Hydrolase, μ-ProtoCadherin, etc.)[21–23]. Ectopic expression of Cdx2 in the mouse stomach triggers the replacement of the gastric mucosae by cells harboring features of intestinal secretory or absorptive lineages[24]. Moreover, Cdx2[−/+] heterozygous mice were found to be more sensitive to tumorigenesis in the distal colon induced either by azoxymethane treatment or Apc mutation[25,26]. It has also been shown that CDX2 positively regulates the expression of the Wnt inhibitors APC and AXIN2[27]. Considering the p400-dependent induction of the Wnt pathway, this suggests a link between CDX2, p400 and thus H2A.Z dynamics to modulate Wnt activity.

Interestingly, a recent study[5] has shown that the global loss of H2A.Z is a signature for differentiation of intestinal stem cells.

Moreover, we recently showed[1] that p400 plays a critical role in colon carcinogenesis through its function in the Wnt pathway. These data underline the potential importance of H2A.Z dynamics in intestinal homeostasis. However, nothing is known about the exact role of H2A.Z in the regulation of gene expression and the set-up of the differentiation-specific transcriptional pattern in this context.

In this study, we investigate the role of H2A.Z dynamics in intestinal epithelial homeostasis. We demonstrate that H2A.Z is essential for the proliferation of intestinal epithelial progenitors and that defects in its expression trigger deregulation of a subset of terminal differentiation marker expressions. We also demonstrate that H2A.Z incorporation into chromatin negatively regulates CDX2 binding to its target genes promoters, thereafter repressing their transcription. Finally, we find that the activity of the Wnt pathway promotes the expression of H2A.Z and impacts the H2A.Z-dependent gene expression. Thus, taken together, we demonstrate an epistatic relationship between histone variant incorporation and binding of the crucial transcription factor CDX2, demonstrating that the H2A.Z histone variant exerts a central role in tissue homeostasis by integrating key signaling by the Wnt pathway.

## Results

**H2A.Z regulates proliferation of intestinal epithelial cells**. We have previously uncovered a major role of some components of the Tip60 chromatin-modifying complex (Tip60 itself and the chromatin remodeler p400) on the proliferation of both normal and colon cancer cells in culture[1,14], and also in vivo, affecting the early steps of colon tumorigenesis[1]. Interestingly, it has recently been shown that the differentiation of intestinal epithelial cells is characterized by a global loss of histone variant H2A.Z[5], a major substrate of both Tip60 and p400. We thus intended to characterize the role of the histone variant H2A.Z in normal intestine homeostasis.

We first analyzed the impact of H2A.Z knockdown on relevant intestinal cell proliferation. We took advantage of the HIEC normal human intestinal crypt cells[28], a unique normal epithelial model harboring features of progenitor cells[28] and which are able to proliferate in culture until confluence without committing into any differentiation process. Transfection of siRNAs directed against H2A.Z mRNA (see knockdown efficiencies in Supplementary Fig. 1) greatly reduces the proliferation of these cells (Fig. 1a), indicating that H2A.Z expression is required for the proliferation of this normal intestinal epithelial crypt cell model.

Similarly, we found that H2A.Z expression is essential for the proliferation of Caco-2/15 cells (Fig. 1b, see Fig. 2a for siRNA efficiency), a human colon cancer-derived cell line. This cell line is largely used to study enterocyte differentiation, which is spontaneously induced when these cells reach confluence. Importantly, in these cells, overexpression of siRNA-resistant H2A.Z reverses, at least in part, the cell proliferation arrest (Supplementary Fig. 2), confirming that the decrease in cell proliferation observed upon H2A.Z siRNA transfection is indeed due to the decrease in H2A.Z expression. Analysis of the mechanism underlying this decrease in cell proliferation does not reveal any induction of apoptosis (assessed by PARP cleavage) nor of markers of senescence (Supplementary Fig. 3). Rather, we observed that depletion of H2A.Z leads to an accumulation of cells in the G0/G1 phase of the cell cycle (Supplementary Fig. 4), suggesting a defect in cell cycle progression.

Thus, all together, these observations indicate that the histone variant H2A.Z is required for the proliferation of both normal and cancer epithelial cells, independently of their differentiation potential.

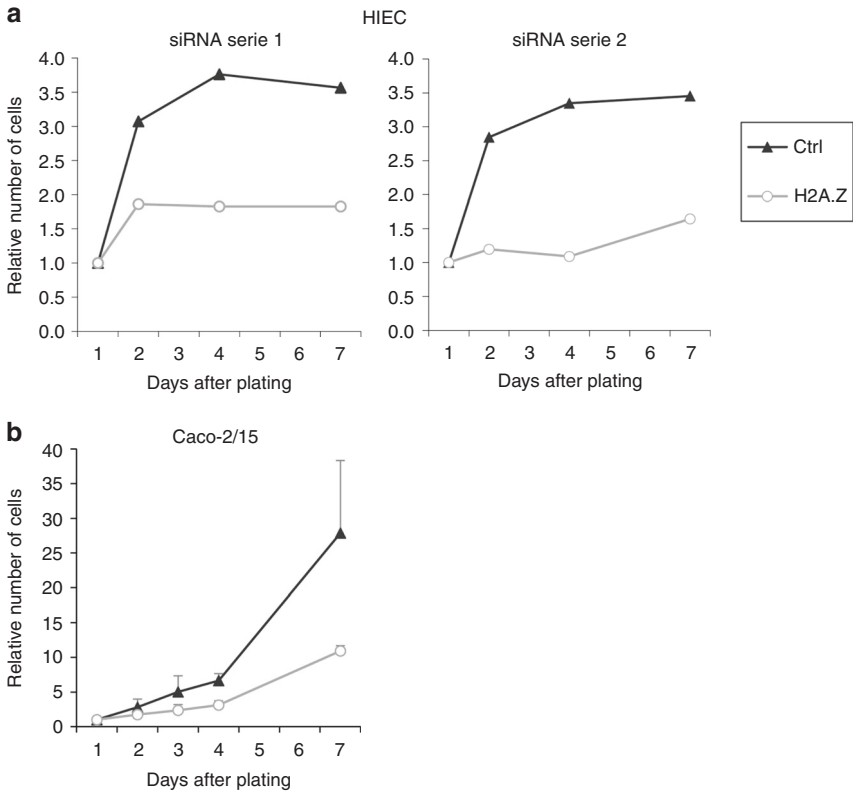

**Fig. 1** H2A.Z-dependent regulation of intestinal epithelial cell proliferation. **a** HIEC cells were transfected using siRNAs targeting H2A.Z mRNA or control. Two different series of siRNAs were used. Cell number was measured and represented relative to 1 for day 1 after transfection and plating. One representative experiment out of two is shown for each serie. **b** Caco-2/15 cells were transfected using siRNAs control or directed against H2A.Z and the cell number was measured and represented relative to 1 for day 1 after transfection and plating. The mean and standard deviation are shown ($n = 3$ independent experiments)

**H2A.Z is a negative regulator of intestine differentiation**. The H2A.Z histone variant has important roles in transcription, inducing positive or negative effects on gene expression depending on chromatin context as well as its post-translational modifications[9].

A very recent study[5] demonstrated that, among chromatin marks that are affected during the intestinal differentiation process, a global genome-wide decrease in H2A.Z incorporation is a signature of Intestinal Stem Cells (ISC) differentiation that correlates with the activation of enterocyte-specific genes. Thus, to investigate the role of H2A.Z in differentiation, we analyzed the impact of H2A.Z down-regulation on the expression of enterocyte differentiation markers. By RT-qPCR and western blot, we observed that, in Caco-2/15 cells, H2A.Z knockdown greatly induced the expression of the terminal enterocyte differentiation marker SI (Fig. 2a), as well as another key protein for enterocyte functions, Lactase Phlorizin Hydrolase (LPH).

Similar results were obtained using another independent siRNA against H2A.Z (Supplementary Fig. 5A), ruling out the possibility of off-target effects. Note that these inductions are much higher than the induction occurring after confluence (in Supplementary Fig. 5B, compare H2A.Z siRNAs to control, cells being seeded at sub-confluence (75%) and Day 0 corresponding to the siRNAs transfection). Importantly, this increase in differentiation markers expression is not due to H2A.Z-depleted cells (Supplementary Fig. 5C) reaching confluence earlier than controls, since proliferation is decreased upon H2A.Z depletion, as shown in Fig. 1b. H2A.Z depletion also results in a weak but significant increase in CDX2 expression (Fig. 2a and Supplementary Fig. 6), which could play a role in the expression of the

differentiation markers (SI and LPH genes being known CDX2 targets). Analysis of other genes bound by H2A.Z[5] revealed an increased expression of KLF4, but not of ARHGEF2 and LDHA (Supplementary Fig. 7), indicating that strong binding of H2A.Z does not determine regulation upon H2A.Z knock-down in Caco-2/15 cells, as already published in other systems[29]. Strikingly, KLF4 is known to be regulated by CDX2[30], which reinforce the link between activation upon H2A.Z depletion and regulation by CDX2.

We next tested the effect of H2A.Z depletion in HIEC2F cells, a non-transformed model derived from HIEC cells. HIEC2F cells express the CDX2 and HNF1α transcription factors in an inducible manner[31], both being important for the differentiation of the intestinal epithelium and for the expression of enterocyte differentiation markers[21]. In the absence of the inducer (Fig. 2b, -dox), HIEC2F cells express CDX2 and HNF1α at moderate levels due to the leakiness of the inducible system (as previously shown by Benoit et al.[31]). We found that, in these non-transformed cells also, depletion of H2A.Z leads to an increase in the expression of differentiation markers SI and LPH (Fig. 2b). This induction requires the presence of CDX2 and HNF1α, since no SI or LPH expression is detected in the parental HIEC wild-type cells which do not express these factors (Benoit et al.[31]). Importantly, in HIEC2F cells, H2A.Z depletion does not induce CDX2 nor HNF1α expression (Fig. 2b). This result indicates that the induction of differentiation markers upon H2A.Z depletion is not mediated by changes in CDX2 and HNF1α expression levels, at least in this cell model. It also suggests that H2A.Z is a direct negative modulator of the expression of the SI or LPH genes.

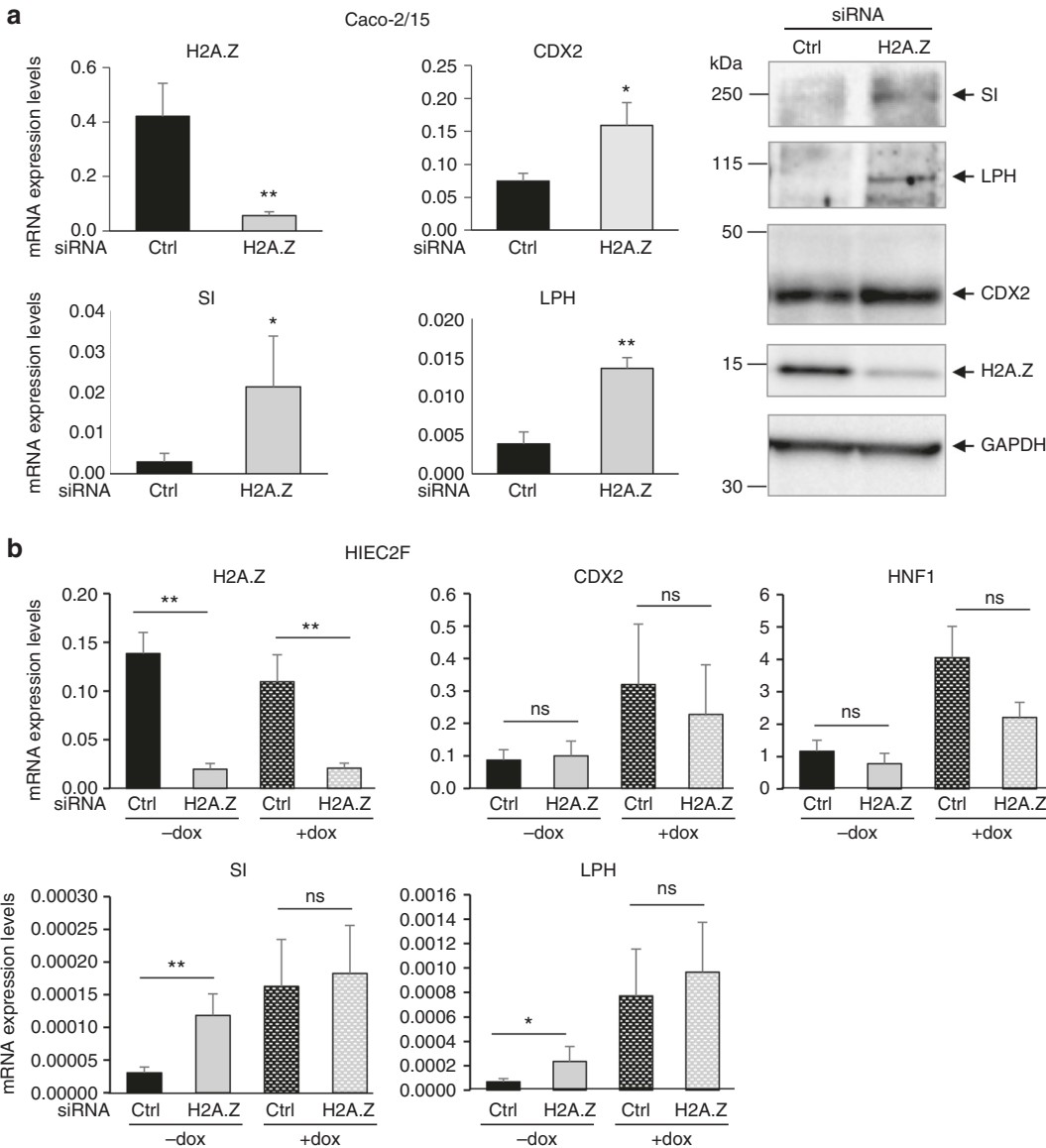

**Fig. 2** Negative control of differentiation marker expression by H2A.Z. **a** Caco-2/15 cells were transfected using siRNA targeting H2A.Z or control. 72 h later, total RNA was extracted and subjected to RT-qPCR analysis for the expression of the indicated genes (calculated relative to β2m mRNA level). The mean and standard deviation are shown ($n = 5$ independent experiments). Statistical analysis was done using the Student's $t$-Test (*$p < 0.05$; **$p < 0.02$ vs control siRNA). Also shown is the expression of the corresponding proteins analyzed by western blot from total extracted proteins of the same samples. **b** HIEC2F cells were transfected using siRNA control or targeting H2A.Z, and treated or not with 10 µg/ml doxycycline for 3 days. Total RNA was extracted and RT-qPCR analysis for the expression of the indicated genes was performed as in **a**

Note that, in the context of the overexpression of CDX2 and HNF1α following doxycycline addition (Fig. 2b, +dox), leading to the induction of enterocyte differentiation markers as previously shown[31], the expression of markers cannot be further increased by H2A.Z knockdown. This absence of effect is probably due to the fact that, when CDX2/HNF1α are strongly overexpressed in the presence of Dox, CDX2/HNF1α -dependent activation of their target genes is maximal and cannot be further increased by H2A.Z depletion. Such a mechanism could suggest a relationship between CDX2/HNF1 activity and H2A.Z effect (see below).

Taken together, these data suggest that H2A.Z acts as a negative regulator of enterocyte differentiation in vitro, both in transformed and non-transformed contexts, by a mechanism dependent on intestine-specific transcription factors.

**H2a.z controls the intestinal epithelial homeostasis in vivo.** We next wondered whether H2A.Z could have the same function in vivo, in the integrated context of the entire organ and organism. We generated a mouse strain allowing the inducible knockout of *H2a.z* in the intestine. We crossed mice homozygously floxed on the *H2afz* gene[32] with the *Lgr5-CRE*[ERT2] mouse strain[33], expressing the CRE recombinase specifically in the intestinal stem cells under the control of the endogenous promoter (heterozygous knock-in) of the intestinal stem cell marker Lgr5. Moreover, the CRE recombinase used in this mouse strain is fused to a modified version of the estrogen receptor ligand binding domain, which sequestrates the enzyme in the cytoplasm in the absence of tamoxifen. Thus, the deletion of the *H2a.z* gene is also temporally controlled and induced by the

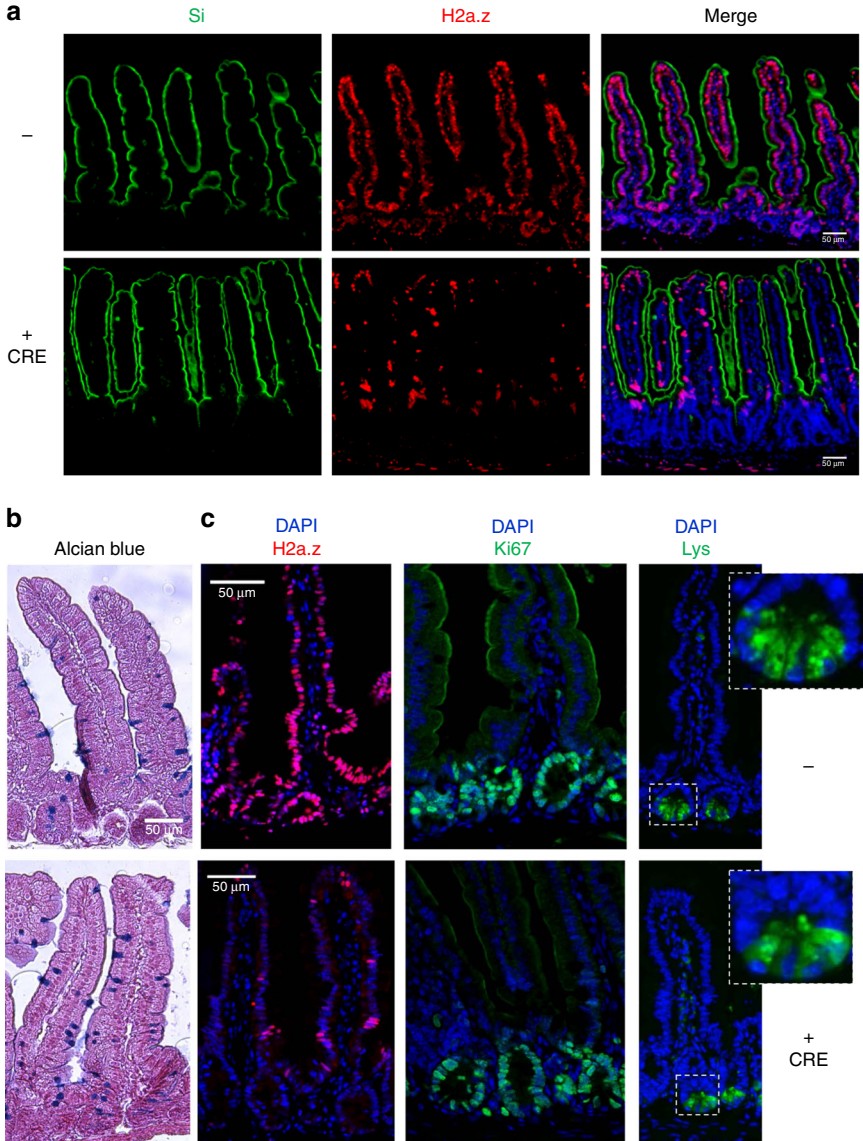

**Fig. 3** Regulation of markers accumulation in tissues upon H2a.z depletion. **a** Immunofluorescence experiments on sections of small intestine from mice expressing (+CRE) or not (−) the CRE recombinase and treated for 10 days with a tamoxifen-containing feed before dissection. The indicated specific antibodies were used, as well as DAPI to stain nuclei (in blue in merge panels), for staining tissues from representative individuals. **b** Alcian blue staining, counterstained with nuclear fast red, was performed on sections, obtained as in **a**, to reveal mucin-producing cells. **c** Same as in **a** using the indicated antibodies and DAPI staining. All panels are representative of the same animals than in **b**

administration of tamoxifen in the food (see Supplementary Fig. 8 for typical genomic recombination efficiency).

We thus obtained an original in vivo model to specifically induce, on demand, the knock-out of H2a.z in intestinal stem cells. Upon tamoxifen treatment, we observed a mosaic disappearance of H2a.z staining as early as 10 days after induction (see central panels of Fig. 3a), in agreement with the fact that LGR5-CRE is known to induce a mosaic knock-out. No obvious change in the size of the crypt-villlus structure was observed (see panels of Fig. 3), nor in the number or the position of the Ki67 positive cells, in the crypts or in the remaining H2a.z -positive cells of the villi (Fig. 3c), suggesting that stem cell maintenance and progenitor cell proliferation was not greatly impaired in vivo. Note however that when we analyzed H2a.z expression one and two months following induction of recombination, we found that knock-out cells were gradually replaced by cells expressing H2a.z (Supplementary Fig. 9),

indicating that H2a.z is required for optimal stem cell maintenance or proliferation.

Interestingly, we observed that *H2a.z* gene deletion leads to a significant increase Sucrase Isomaltase (Si) expression both at the protein and mRNA levels (Figs. 3a and 4a), providing evidence that, as previously observed in cell culture, *H2a.z* depletion also induces some enterocyte-specific markers in vivo. Note, however, that *Lph* gene expression is not significantly affected by H2a.z knock-out (Fig. 4a), perhaps because regulatory mechanisms are species- or context-specific. Alternatively, the weak phenotype of these mice when analyzing mRNA expression could be due to the variegated incomplete recombination of the *H2a.z* gene we describe above.

Interestingly, deregulating H2A.Z levels also leads to an increase in the number of goblet cells (another important epithelial lineage of the mature intestine) all along the villi (Fig. 3b). At the mRNA level, we found that messengers encoding

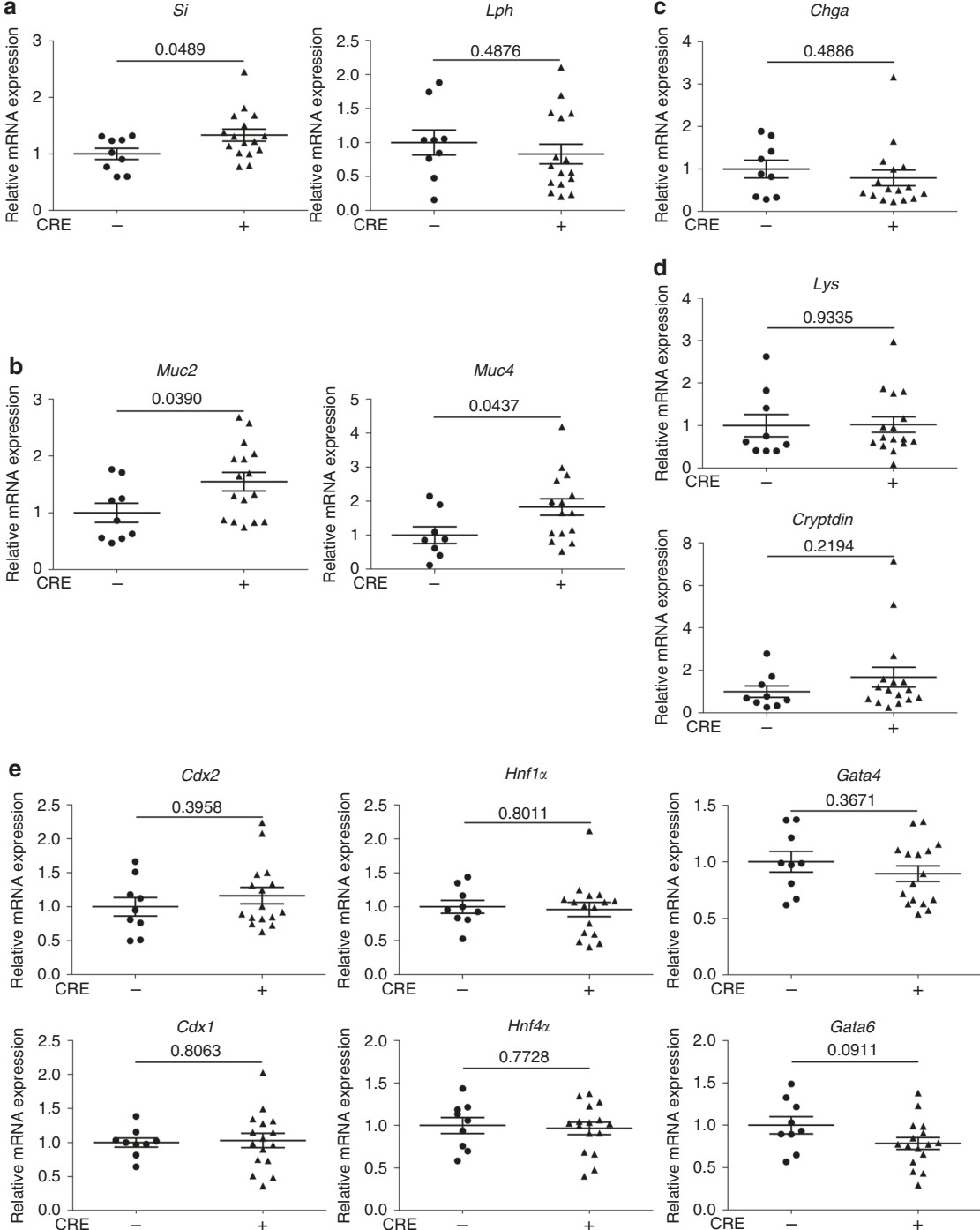

**Fig. 4** H2a.z controls the expression of differentiation marker mRNAs in vivo. 10–15 mice per indicated genotypes were fed a tamoxifen-containing diet for 10 days, sacrificed and the small intestinal mucosae were harvested and total RNA extracted, as indicated in the Materials and Methods. qPCR was then performed for the detection of enterocyte (**a**), goblet (**b**), entero-endocrine (**c**), and Paneth (**d**) cell markers. β2m-normalized values for each mouse were plotted relative to 1 for the mean of control (-CRE) mice. Means and standard errors are represented and the Student's *t*-Test *p*-value is indicated for each gene. **e** Same as in **a–d** for intestine-specific transcription factors

the Muc2 and Muc4 proteins, two differentiation markers for goblet cells are also induced upon H2a.z knock-out (Fig. 4b).

Differentiation markers for entero-endocrine cells (Chromogranin A) (Fig. 4c) or Paneth cells (Cryptdin or Lysozyme) (Figs. 4d and 3c) are unchanged whatever the expression level of H2a.z.

Altogether, these data suggest that H2a.z inhibits the expression of some differentiation markers of enterocyte and goblet cell lineages in the intestinal epithelium in vivo.

**H2A.Z inhibits CDX2 binding to its target promoters**. We next intended to characterize the mechanism by which H2A.Z regulates the expression of intestinal differentiation markers.

Analysis of the mRNA expression of intestine-specific transcription factors (Fig. 4e) does not reveal any obvious changes upon H2a.z depletion in vivo, suggesting that the effect of H2A.Z on differentiation markers is not mediated by the regulation of the expression of such transcription factors.

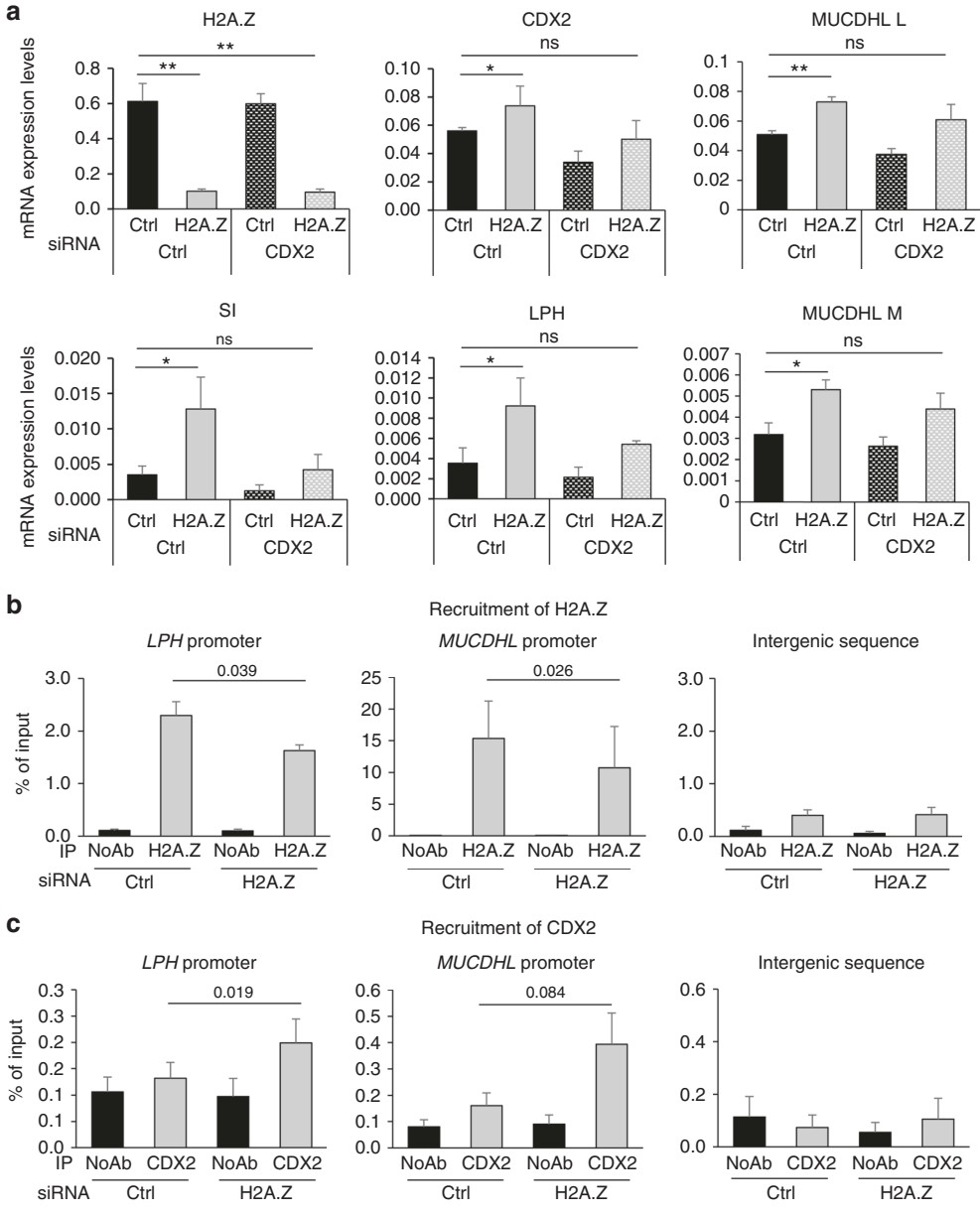

**Fig. 5** Epistatic relationship between H2A.Z and CDX2. **a** Caco-2/15 cells were transfected using siRNA targeting CDX2 mRNA or control, upon or not siRNA-mediated H2A.Z depletion. 72 h later, total RNA was extracted and subjected to RT-qPCR analysis for the expression of the indicated genes (calculated relative to β2m mRNA level). The mean and standard error are shown ($n = 3$ independent experiments). Statistical analysis was done using Student's $t$-Test (*$p < 0.05$; **$p < 0.02$ vs control siRNA). **b** Caco-2/15 cells were transfected using siRNA targeting H2A.Z or control, as indicated. 72 h later, ChIPs were performed using an H2A.Z antibody or no antibody as a control (No Ab). The amount of *LPH* or *MUCDHL* promoters, or an intergenic sequence were analyzed by qPCR, and the percentage of input was calculated. The mean and standard error are shown ($n = 3$ independent experiments). Statistical analysis was done using unilateral Student's $t$-Test. **c** Same in **b** for CDX2 ChIP samples

Strikingly however, CDX2 is a critical transcription factor for the two cell lineages (enterocytes and goblet cells) which we found affected by H2A.Z depletion and is known to directly regulate the genes induced by H2A.Z depletion in Caco2/15 cells. We thus postulated that CDX2 could mediate the effect of H2A.Z on the expression of these genes. Such an epistatic relationship between CDX2 and the H2A.Z would be consistent with our findings from Fig. 2b that no effect of H2A.Z depletion could be observed upon strong overexpression of CDX2.

In order to test whether H2A.Z effects are dependent on CDX2, we co-transfected Caco-2/15 cells with siRNAs directed against H2A.Z and CDX2 alone or in combination (Fig. 5a). CDX2 siRNA indeed induces a detectable decrease in CDX2 expression at the mRNA and protein levels (Fig. 5a and Supplementary Fig. 6), as well as its target genes (Fig. 5a). We observed that the induction of SI and LPH mRNA resulting from H2A.Z knockdown is largely reversed by the concomitant reduction of CDX2 expression. Similarly, the expression of μ-protocadherin (MUCDHL spliced isoforms L and M), an adhesion molecule known to be expressed by differentiated colorectal epithelial cells, is also found to be upregulated upon H2A.Z knockdown in a CDX2-dependent fashion.

This result thus indicates that the effects of H2A.Z knock-down on the expression of enterocyte-specific genes are dependent on CDX2 activity.

CDX2 directly binds to the promoters of differentiation-specific genes[21,34,35]. We thus tested whether H2A.Z could regulate the binding of CDX2 to its target promoters.

We transfected Caco-2/15 cells with a siRNA directed against H2A.Z and analyzed H2A.Z presence (Fig. 5b) and CDX2 recruitment (Fig. 5c) to the promoters of CDX2 target genes by ChIP. H2A.Z depletion leads to a weak but significant decrease in the presence of H2A.Z-containing nucleosomes at the *LPH* and *MUCDHL* promoters (Fig. 5b), indicating that the global decrease of H2A.Z levels induced by siRNAs translates into a local decrease in H2A.Z genomic occupancy at these promoters. For unclear reasons, we were unable to detect significant ChIP signals at the SI promoter, even using previously published primers and with total histone H3 antibodies.

Strikingly, upon H2A.Z knockdown (which induced markers expression as expected (Supplementary Fig. 10), we observed a specific increase in the recruitment of CDX2 to these two promoters (Fig. 5c), as well as to the KLF4 promoter (Supplementary Fig. 7B), which, as shown above, was also activated upon H2A.Z depletion (Supplementary Fig. 7A).

Thus, this result suggests that reducing H2A.Z at the promoters reduces the chromatin recruitment of the CDX2 intestine-specific transcription factor, at least for its targets *LPH* and *MUCDHL*.

**H2A.Z integrates the Wnt-dependent control of homeostasis**. The results described above indicate that artificial modulations of H2A.Z expression can affect intestinal homeostasis. We next tested whether the molecular circuitry we uncovered was targeted by signaling pathways that control intestinal homeostasis.

Interestingly, we previously uncovered a link between the H2A.Z-incorporating p400 ATPase and the Wnt pathway. We thus intended to analyze the impact of Wnt pathway activity on H2A.Z expression. We transfected into Caco-2/15 cells a siRNA directed against β-catenin (a central actor of the Wnt pathway). As expected, transfection of this siRNA reduced the expression of β-Catenin (*CTNNB1*) mRNA (Fig. 6a), as well as Wnt target genes (Fig. 6b). Strikingly, we found that this depletion also decreased the expression of H2A.Z, while SI, LPH MUCDHL and KLF4 expression increased (Fig. 6c and Supplementary Fig. 7C). The same effects were observed using another β-Catenin-targeting siRNA (Supplementary Fig. 11A) or FH535 Wnt pathway inhibitor (Supplementary Fig. 11B). These results provide evidence that Wnt/β-Catenin signaling favours the expression of H2A.Z and concomitantly represses the expression of differentiation markers.

In order to test whether the regulation of H2A.Z expression by the Wnt/β-Catenin/TCF pathway could be direct, we analyzed published TCF7L2(TCF4) ChIP-Seq data[36]. Interestingly, a strong TCF4 signal was observed around the H2A.Z promoter in HCT116 cells (Fig. 7a), which are derived from colorectal cancer, as Caco2 cells: a major TCF4 peak could be seen about 500 bp upstream from the H2A.Z transcription start site (TSS), whereas weaker binding was observed downstream from the TSS. Strikingly, no binding could be observed in cells originating from other tissues whereas the binding on the well-known target gene *LGR5* was observed in all cell types (Fig. 7a). Importantly, the regulation of H2A.Z expression by the Wnt pathway was conserved in HCT116 cells, as in Caco-2/15 cells, since depletion of β-Catenin also induced a significant decrease in H2A.Z mRNA expression (Fig. 7b). We confirmed by ChIP-qPCR experiments that TCF4 binds to the H2A.Z promoter in HCT116 cells, and also observed significant binding at the H2A.Z promoter in Caco2 cells (Fig. 7c).

In order to show that the regulation of H2A.Z expression by β-Catenin siRNA is indeed due to the binding of TCF4 to the H2A.Z promoter, we used CRISPR-mediated genome editing to remove the TCF4 strong binding site in HCT116 cells (see Supplementary Fig. 12 for characterization of the cell line). ChIP experiments indeed showed that the binding of TCF4 to the H2A.Z promoter was decreased in genome edited cells compared to parental cells (Fig. 8a), although not abolished presumably because of the second TCF4 binding site present downstream from the TSS. We next transfected genome-edited cells and parental cells using siRNA directed against β-Catenin. We found that knock-down efficiency was similar in both cell lines, resulting in a similar decrease in the Wnt target genes *LGR5* and *CCND1* (Fig. 8b). However, the decrease in H2A.Z expression is significantly lower in the genome-edited cells compared to control cells (Fig. 8b), the residual decrease being probably due to the residual binding observed in TCF4 ChIP experiments. Altogether, these data indicate that the H2A.Z promoter is a direct target of TCF4/β-Catenin, the main transcriptional effectors of the classical Wnt pathway.

Interestingly, we further found that the upregulation of differentiation markers expression we observed upon β-Catenin knock-down was counteracted by the concomitant overexpression of H2A.Z, at least in part (Fig. 9a). This effect does not rely on an indirect effect of H2A.Z overexpression on Wnt signaling, since proliferative Wnt target genes, such as *LGR5* or *CCND*1, are not affected by H2A.Z overexpression (Fig. 9b).

Taken together, our data show that the activation of H2A.Z expression is one of the mechanisms by which the Wnt signaling pathway controls progenitor maintenance and differentiation of the intestinal epithelium. As a consequence, they point towards a central integrative role of H2A.Z in the control of intestinal homeostasis (Fig. 9c).

## Discussion

In this work, we demonstrate the functional importance of H2A.Z expression in preventing terminal differentiation into various lineages of intestinal progenitor cells. To our knowledge, this is the first demonstration of such a causal role of H2A.Z in vivo.

We propose a model in which the presence of H2A.Z impairs the binding of the intestine specific transcription factor CDX2 to its target promoters. The first question raised by our finding is about the mechanism by which H2A.Z prevents CDX2 binding. In vitro experiments using recombinant nucleosomes indicate that both H2A- and H2A.Z-containing nucleosomes block binding of CDX2 to its target site (Supplementary Fig. 13). These data show that it is not by merely replacing canonical H2A that H2A.Z blocks CDX2 binding.

We can thus envision some non-exclusive possibilities. First, the depletion of H2A.Z upon siRNA or upon differentiation could lead to the decrease of local nucleosome occupancy, resulting in higher binding of CDX2 to its target sites. Interestingly and in agreement with this possibility, we found that H2A.Z depletion induces a significant reduction in H3 enrichment around the TSS of differentiation markers genes (Supplementary Fig. 14). Thus, the eviction of nucleosomes in the CDX2 targeted regions of promoters could be, at least in part, involved in the induction of these genes upon decrease of H2A.Z levels.

However, we can also speculate that post-translational modifications of H2A.Z or of the H2A molecule, which replaces it could be involved in regulating the recruitment of CDX2. H2A.Z has indeed been shown to be modified by acetylation or sumoylation, and one of these modifications could directly or indirectly (through the recruitment of specific proteins) prevent CDX2 binding. Alternatively, the histone H2A that replaces H2A.Z could be modified in such a way that this modification favours CDX2 binding. Investigating these possibilities clearly opens a

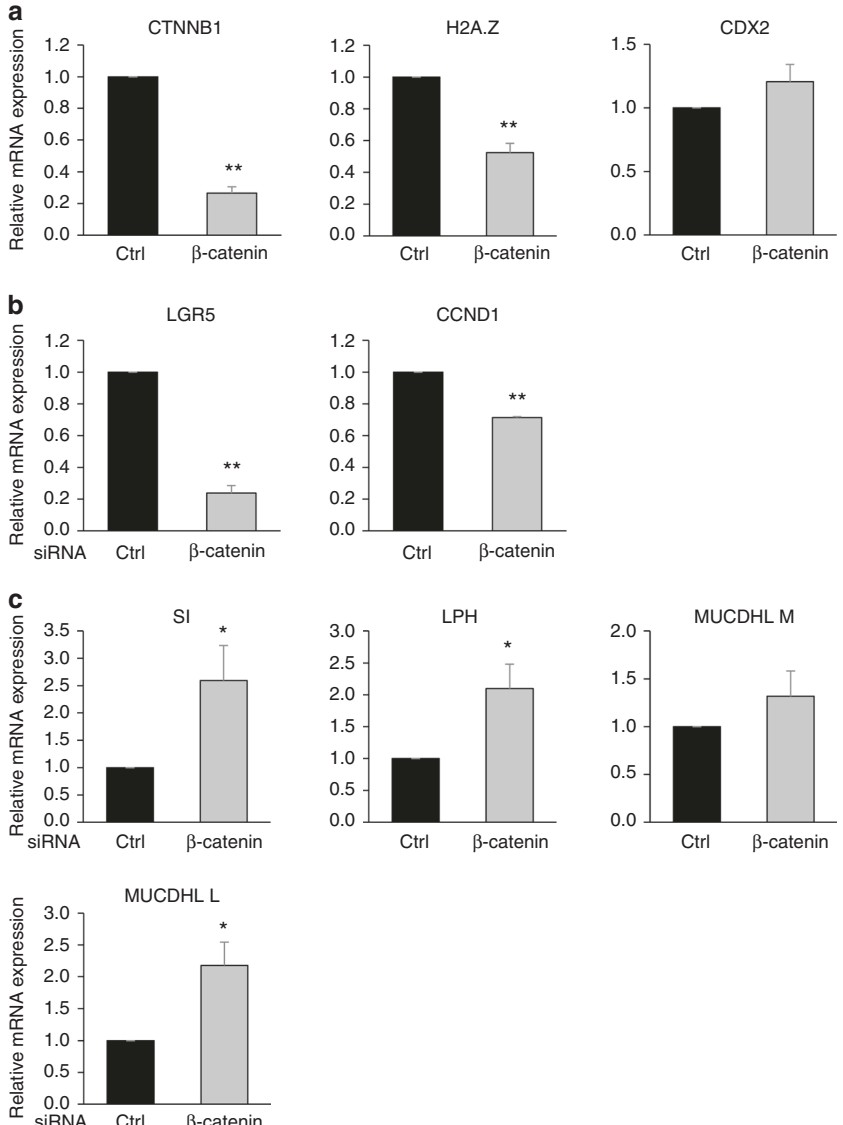

**Fig. 6** β-catenin positively regulates H2A.Z and impairs differentiation marker expression. **a** Caco-2/15 cells were transfected using siRNA targeting β-catenin mRNA or control. 72 h later, total RNA was extracted and subjected to RT-qPCR analysis for expression of the indicated genes (calculated relative to β2 M and RPLP0 (p0) mRNA levels). The mean and standard errors are shown ($n = 4$ independent experiments). Statistical analysis was done using Student's $t$-Test (**$p < 0.02$ vs control siRNA). **b** Same as in **a** for Wnt target genes (**$p < 0.02$ vs control siRNA). **c** Same as in **a** for enterocytes differentiation marker genes (*$p < 0.05$ vs control siRNA)

new research avenue on the regulation of CDX2 binding to its target genes and the consequent control of intestine homeostasis.

Our study allows us to propose that the role of H2A.Z on the differentiation process is probably to avoid the ectopic expression of differentiation markers in the crypts. Interestingly, a recent work of Kazakevych et al.[5] uncovered a strong correlation between intestinal stem cell differentiation and a reduction of H2A.Z expression levels. Moreover, previous works have shown that the Tip60/p400 complex, which is a critical actor in H2A.Z dynamics, being involved both in H2A.Z incorporation into and removal from the chromatin, is involved in the renewal of normal ES cells[37]. In these cells, the knockdown of some subunits of the complex induces a decreased proliferation rate, the reduction of stemness abilities and the ectopic expression of differentiation markers.

Together with these findings, our data demonstrate that H2A.Z dynamics is a major determinant of cell fate control in an integrated model. Although it has already been shown that other chromatin marks can be determinant for stemness maintenance or the induction of differentiation in the context of the intestinal epithelium[38,39], our work demonstrates that a structural chromatin-related feature, i.e. H2A.Z incorporation, serves as a key event in the maintenance of progenitors in the intestinal epithelium.

Whether H2A.Z plays a similar role in other tissues or contexts is clearly an open issue. The role of the Tip60/p400/H2A.Z complex in the renewal of normal ES cells described above is consistent with a general role for H2A.Z in preventing the expression of differentiation markers. However, some recent data suggest that this role cannot be generalized to all tissues. Indeed, Shen et al.[32] describe the exact opposite role for H2A.Z in neurogenesis since they demonstrated that the depletion of H2A.Z in transgenic mice enhances the proliferation of neural progenitors and triggers important defects in neuron differentiation, as well as in learning and memory. It has also been shown[40] that the depletion of H2A.Z in MDCK canine kidney epithelial cells can

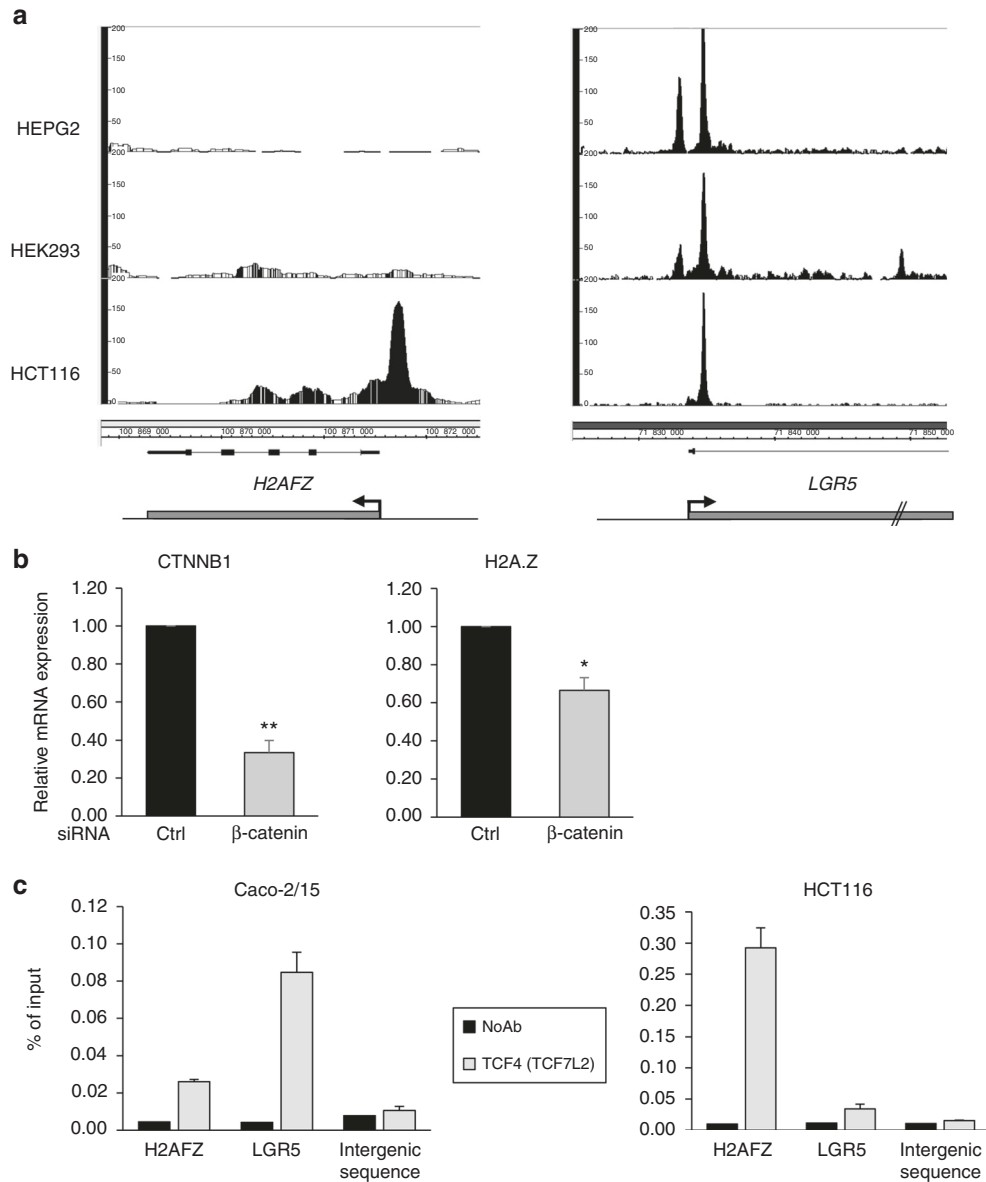

**Fig. 7** H2A.Z is a target of the Wnt pathway. **a** Analysis of ChIP-seq data from Frietze et al.[36] for the binding of TCF7L2 on *H2AFZ* and *LGR5* gene promoters in three cells lines (HEPG2, HEK293, and HCT116). Representation of peaks of enrichment were obtained using Integrated Genome Browser 9.0.2 (BioViz, https://bioviz.org). Note that the drawing scale is not identical due to the relative sizes of both genes. **b** HCT116 cells were transfected using β-catenin –targeting or control siRNA. 72 h later, total RNA was extracted and subjected to RT-qPCR analysis for expression of the indicated genes (calculated relative to β2 M and RPLP0 (p0) mRNA level). The mean and standard errors are shown ($n = 3$ independent experiments). Statistical analysis was done using Student's *t*-Test (*$p < 0.05$; **$p < 0.02$ vs control siRNA). **c** Chromatin from sub-confluent HCT116 and Caco-2/15 cells were harvested and ChIPs were performed using a TCF7L2(TCF4) antibody. The amount of *H2AFZ* or *LGR5* promoters, or an intergenic sequence were analyzed by qPCR, and the percentage of input was calculated. The mean and standard errors are shown ($n = 3$ technical replicates)

favor epithelial-mesenchymal transition (EMT) and promote the invasive properties of these cells. On the contrary, and in accordance with our work, it has been demonstrated that H2A.Z can exhibit an oncogenic potential and can be required for the EMT in hepatocellular carcinoma[41]. Thus, H2A.Z is probably widely involved in tissue homeostasis and in the regulation of key physiological functions, but its biological role is organ-specific and probably dependent on the local context (chromatin landscape, proteins recruited to chromatin, etc.).

Our data also provides clues on the relationship between the Wnt signaling pathway and intestinal homeostasis. The data presented here show that Wnt signaling favours H2A.Z expression and, since we found that H2A.Z prevents CDX2 binding to promoters, this pathway also regulates the expression of CDX2-target genes. Thus, we found that some of these genes, associated with enterocyte differentiation, such as *Sucrase-Isomaltase* or *MUCDHL*, are repressed by Wnt signaling. We also demonstrated the causative role of H2A.Z repression in the effects observed upon β-Catenin knock-down. These data indicate that the control of H2A.Z expression is an important molecular event by which the Wnt signaling pathway controls intestinal homeostasis. Whether H2A.Z could also play such a role outside of intestinal homeostasis is clearly worth investigating.

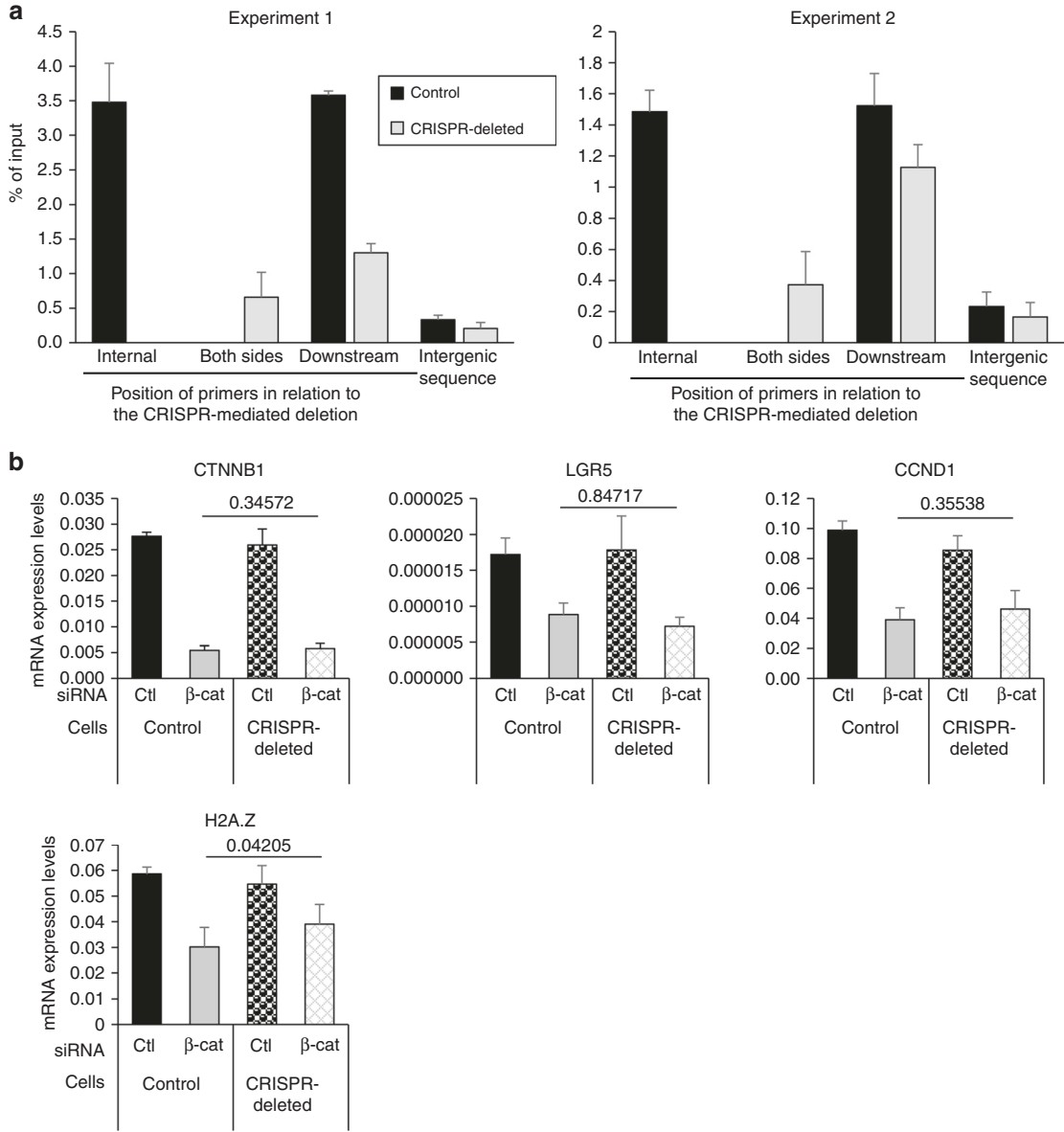

**Fig. 8** H2A.Z is an effector of the Wnt pathway on differentiation markers. **a** Chromatin from sub-confluent CRISPR-Cas9-modified HCT116 cells to homozygously delete the major TCF7L2 binding region of the *H2AFZ* gene promoter, using indicated *H2AFZ* promoter—specific or intergenic sequence—targeting primers. Results and standard errors ($n = 3$ technical replicates) are shown for two independent experiments. **b** Control or CRISPR-modified HCT116 cells were transfected using β-catenin—targeting or control siRNA. 72 h later, total RNA was extracted and subjected to RT-qPCR analysis for expression of the indicated genes (calculated relative to GAPDH and β-actin mRNA levels). The mean and standard errors are shown ($n = 4$ independent experiments). The Student's *t*-Test *p*-value was calculated (for β-catenin -depleted samples from control vs CRISPR-modified cells) and is indicated for each gene

Finally, it has been shown that MUCDHL, which we found to be induced upon H2A.Z knockdown in Caco-2/15 cells, is a CDX2 target gene and an important modulator of Wnt-dependent homeostasis[23]. Indeed, it acts as a negative Wnt regulator, by sequestering β-Catenin at the plasma membrane, and its expression is also inhibited by Wnt signaling[42]. Thus, H2A.Z could control, at least in part, Wnt-dependent proliferation by the same mechanism as differentiation, i.e. by regulating CDX2 binding to the *MUCDHL* promoter. This would indicate that H2A.Z can also be upstream of Wnt signaling, consistent with our previous finding[1] that p400 is a crucial regulator of the Wnt signaling pathway. Altogether, these data indicate that H2A.Z dynamics and Wnt signaling are strongly interconnected, raising the possibility that other Wnt-dependent processes are also controlled by H2A.Z dynamics and its regulators.

## Methods

**Ethics statement**. The experiments involving animals were conducted according to French governmental norms. Authors have complied with all relevant ethical regulations for animal testing and research. This study was approved by the Ethics Committee of the institute "Centre de Biologie Intégrative" (FR3743) and was authorized by the French Ministry of Education and Research (approval APAFIS #4528-2016031109479615 v3). We have complied with all relevant ethical regulations.

**Animals**. Mice homozygously floxed on the *H2afz* and *H2afv* genes were obtained from RIKEN BRC (Ibaraki, Japan) and back-crossed with C57Bl/6 J mice (Charles-River, L'Arbresle, France). The F1 offspring were then crossed with each other to generate *H2afz*[fl/fl] mice. These mice were then crossed with the *Lgr5-Cre*[ERT2] mouse model (from Jackson Laboratory) to obtain F1 littermates also crossed with each other to obtain *Lgr5-Cre*[ERT2]/*H2afz*[fl/fl] mice. Genotyping for *H2afz* and *Cre* alleles was done by PCR analysis of tail DNA samples (see Supplementary Table 1 for primer sequences).

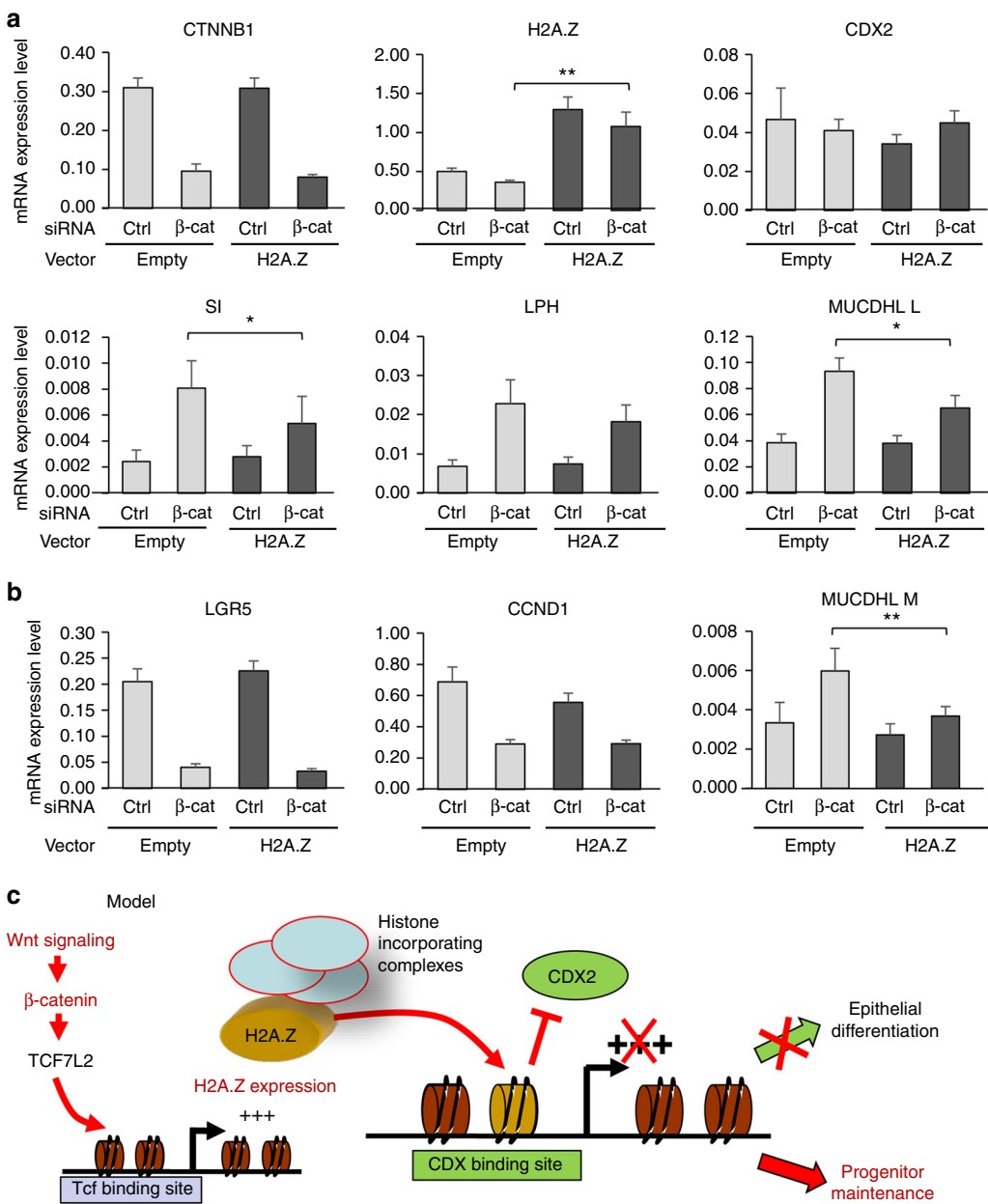

**Fig. 9** H2A.Z is an effector of the Wnt pathway on differentiation markers. **a** Caco-2/15 were treated using siRNA targeting β-catenin mRNA or control and transfected with vectors encoding or not H2A.Z to analyse the impact of H2A.Z forced expression on the effects of β-catenin knockdown. The expression of the indicated genes was measured for each condition (calculated relative to β2 M and RPLP0 (p0) mRNA levels). The mean and standard error are shown ($n = 6$ independent experiments). Statistical analysis was done using Student's $t$-Test (*$p < 0.05$; **$p < 0.02$ vs β-catenin siRNA + empty vector condition). **b** Same as in **a** for Wnt target genes. **c** Working model: in our study, we showed that the Wnt/β-catenin/TCF7L2(TCF4) signaling pathway positively regulates the expression of the H2A.Z histone variant. The incorporation of the variant, mediated by chromatin modifying complexes, is then increased and responsible for the low recruitment of the intestine-specific transcription factor CDX2 on its target genes and thus, limits the expression of such genes. Since it is known that H2A.Z occupancy of a specific promoter is a signature for intestinal stem cells and that this enrichment is reduced upon differentiation, our data show that H2A.Z participates in the regulation of epithelial differentiation through Wnt signaling. Thus, our study demonstrates that a structural chromatin mark can control the cell fate of normal progenitors and, thereafter, intestinal epithelial homeostasis

In experiments, 6–8-week-old male mice were fed a Tamoxifen-containing diet (Tam Diet TD130857 (500 mg/kg), ENVIGO, USA) for ten days. The administration route and dosage had previously been studied and validated[43]. During treatment, macro-physiological parameters (weight, behavior, activity, aspect, etc.) of mice were monitored daily.

**Tissue sampling**. After treatment, the mice were euthanized, dissected and the second third of the small intestine (jejunum) was opened longitudinally. Then, cells were harvested by scraping the mucosae using a scalpel blade, before being subjected to DNA, RNA or protein extraction. The proximal part of each tissue sample was fixed in formalin before being paraffin-embedded, sectioned and used in immunofluorescence experiments.

**Tissue sections and immunofluorescence**. Fixed intestines were dehydrated and embedded in paraffin before being cut into 5 μm sections. For immuno-fluorescence, tissue sections were deparaffinized with Histo-clear II (Euromedex) and then rehydrated in successive baths of ethanol (100, 95, and 70%) and distilled water. The slides were heated in unmasking solution (Eurobio) with a pressure cooker to reveal antigens. Then the tissue sections were permeabilized for 5 min with Triton X-100 1% and blocked with BSA 1% for 45 min. The sections were incubated with primary antibodies (diluted 1:500) for 2 h at room temperature. The

following antibodies were used: H2A.Z (ab4174, Abcam), SI (sc-393470, Santa-Cruz), Lysozyme (ab108508, Abcam) and Ki67 (ab15580, Abcam). The sections were incubated with secondary antibodies (dilution 1:500) coupled with Alexa Fluor® (Fisher Life Science, anti-mouse -AF488 #A21202 and -AF594 #A11032; anti-rabbit -AF488 #A11034 and -AF594 #A11037) for 1 h at room temperature, and the nuclei were stained with DAPI for 3 min at room temperature.

For Alcian Blue staining, rehydrated sections were incubated with acetic acid 3% for 3 min at room temperature before incubation with Alcian Blue (1% in acetic acid, pH 2.5, Sigma-Aldrich) for 30 min. Then the sections were counterstained with nuclear fast red (0.1%, 5 min, Sigma-Aldrich), dehydrated in ethanol (70%, 95%, 100%) and clarified in Histo-clear II. Finally the slides were mounted with ProLong® (Fisher life Science) and stored at 4 °C.

**Cell culture and treatments**. The colorectal cancer cell lines Caco-2/15 and HCT116, and the human intestinal crypt cell line HIEC, were cultured in Dulbecco's Modified Eagle's Medium (DMEM). The Caco-2/15 and HCT116 cells were passed every 3 days at sub-confluence. Cells were transfected with siRNA (see Supplementary Table 2 for sequences) or an H2A.Z expression vector (kind gift from Peter Cheung, University of Toronto, ON Canada) by electroporation (Amaxa) and analysis was performed 3 days after transfection. For proliferation assays, transfected cells were seeded in 96-well plates (2000 cells per well). Then at each day of the kinetics, cells were incubated with the cell proliferation reagent WST-1 (Sigma) for 2 h at 37 °C, and cell numbers were measured by optical density at 450 nm.

Caco-2/15 and HIEC cells were from the Jean-François Beaulieu's lab (Université de Sherbrooke, Québec, Canada) and HCT116 were obtained from Patrick Calsou's lab (IPBS, Toulouse, France).

**RT-qPCR and western blot analysis**. Genotyping for *H2afz* alleles in intestinal epithelial samples was done by PCR analysis of DNA samples (see Supplementary Table 1 for primer sequences).

For RT-qPCR, the RNA from cells or tissues was extracted using the RNeasy Kit (Qiagen) according to the manufacturer's protocol, then it was reverse transcribed into cDNA using AMV reverse transcriptase (Promega). Finally quantitative PCR was performed using specific primers (see Supplementary Tables 3 and 4), and β2m was used as housekeeping gene.

For western-blot, the proteins were extracted in lysis buffer (Triton X100 1%, SDS 2%, NaCl 150 mM, NaOrthovanadate 200 μM, Tris/HCl 50 mM). The proteins were separated in NuPage BisTris 4–12% gels (Invitrogen), then transferred onto PVDF membranes. These membranes were incubated with primary antibodies (diluted 1:500): SI (Novus, NBP1–62362), LPH (Biorbyt, orb184881), CDX2 (Abcam, ab88129), H2A.Z (Abcam, ab4174), PARP (Cell Signalling, 9542), CDKN2A$^{P16INK4a}$ (Abcam, ab108349) and GAPDH (Chemicon, Mab374). Finally, they were incubated with secondary antibodies (dilution 1:500) coupled with HRP (BioRad, anti-mouse-HRP #1706516 and anti-rabbit-HRP #1706515), and revealed with Lumi-light$^{PLUS}$ substrate (Roche). Original uncropped and unprocessed images are presented in Supplementary Fig. 15.

**Chromatin immunoprecipitation**. ChIP experiments were performed classically[44]. Cells were fixed with formaldehyde 1% for 15 min, then cells and nuclei were lysed. The recovered chromatin was sonicated: 10 cycles of 10 s (1 s on, 1 s off) and precleared. Fifty micrograms for H2A.Z and 200 μg for CDX2 of chromatin were used for immunoprecipitation with 2 μg of antibodies (H2A.Z: Abcam, ab4174; CDX2: Bethyl, A300–691A; TCF7L2: Cell signaling, C48H11; Histone H3: Abcam, ab1791). Then, the chromatin was incubated with A/G beads for 2 h. Crosslinking was reversed by incubation of the beads with SDS at 70 °C and proteins were degraded with proteinase K. Finally, DNA was purified using the GFX™ DNA purification kit (GE Healthcare), and ChIP was analyzed by qPCR using specific primers (see Supplementary Table 5).

**CRISPR-Cas9 -mediated genome editing**. HCT116 cells were made defective for the TCF7L2 binding region on the H2A.Z promoter by using CRISPR-Cas9 technology adapted from Agudelo et al.[45]. Briefly, cells were transfected using JetPEI (Polyplus) and plasmids encoding for Cas9, PAM and guide for NA/K ATPase and PAM sequences allowing the targeting of the TCF7L2 peak on the H2A.Z promoter (Supplementary Table 6). Three days after transfection, 0.4 μM ouabain was added for 72 h to select recombinant resistant clones and homozygote knock-out cell lines were screened by sequencing.

**Cycle cycle analysis by EdU staining**. Caco-2/15 cells were transfected using siRNA directed against H2A.Z. Three days later, cells were treated with 10 μM of EdU for 2 h. Cells were prepared for cell cycle analysis using Click-iT™ EdU Kit (ThermoFisher) according to the manufacturer's protocol. Briefly, cells were recovered with trypsin and treated with Click-iT™ Fixative for 15 min. Fixed cells were permeabilized using Click-iT™ saponin -based reagent, and incubated with the Click-iT™ Reaction cocktail (CuSO4 100 mM, AlexaFluor 488), for 30 min at room temperature, to detect EdU. Cells were then incubated with Propidium iodide (BD Biosciences) for 30 min at 37 °C to stain DNA. EdU and PI stainings were

measured using FACSCalibur™ (BD Biosciences) and data were analyzed with FlowJo software.

**In vitro binding of CDX2 to nucleosomes**. DNA fragments corresponding to proximal *SI* and *LPH* promoters, of 150pb and 156pb respectively, were generated by PCR on HIEC genomic DNA using Cy5-labelled forwardprimers (Supplementary Table 7).

To generate mononucleosomes containing H2A or the H2A.Z variant, H3-H4 tetramers and H2A-H2B or H2A.Z-H2B dimers (1 μg/μl each, Diagenode) were assembled with labelled PCR products, in a histones/DNA weight ratio of 0.2 in a solution containing 2 M NaCl and 10 mM Tris-HCl (pH7.5), and incubated 10 min at 37 °C. Then, the mixture was diluted at 0.5 M NaCl and incubated 30 min at the same temperature and finally dialysed at 4 °C against 10 mM Tris-HCl (pH7.5) + 1 mM EDTA for 2 h.

Recombinant His-tagged CDX2 protein was expressed in *E.coli* and purified on nickel-NTA resin.

Labelled PCR products from *SI* or *LPH* promoters, free or integrated in nucleosomes, were then incubated with or without CDX2 recombinant protein for 30 min at room temperature in 20 mM Hepes (pH7.5), 100 mM NaCl, 1 mM EDTA, 1 mM DTT and carrier DNA.

For gel shift mobility assays, 4% polyacrylamide (acrylamide-bisacrylamide, 29:1 [wt/wt], BioRad) slab gels in TBE buffer were used. After a 1 h pre-run at 200 V, samples were loaded and electrophoresis was then performed at the same voltage for 3 h at room temperature. Then, gels were analysed using Typhoon device (Perkin Elmer).

**Reporting summary**. Further information on research design is available in the Nature Research Reporting Summary linked to this article.

## Data availability

The authors declare that all data supporting the findings of this study are available within the paper and its supplementary information files or from the corresponding author upon request.

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

## Acknowledgements

We thank Elizabeth Herring for reviewing the manuscript, Cécile Coiffait and Rémi Boudou for their help in the set-up of the in vivo and preliminary experiments, and also Dr Violette Morales (CBI—LMGM) and Dr Yvan Canitrot (CBI-LBCMCP) for their technical support in nucleosome mobility assays and CRISPR-Cas9 experiments, respectively. We thank the ABC animal facility of the CBI and ANEXPLO for housing the mice. Microscopy experiments were performed at the Toulouse Réseau Imagerie TRI facility located at the Centre for Integrative Biology (CBI) of Toulouse. This work was supported by a grant from the Fondation ARC as a "Programme ARC" to DT. JR is recipient of a studentship from the French Ministry of Research.

## Author contributions

J.R., L.B., M.C.-B., F.E.: Acquisition of data, analysis and interpretation of data. J-F.B.: Technical and material support. J-F.B., M.C-B., D.T., F.E.: Critical revision of the manuscript for important intellectual content. J.R., F.E.: Statistical analysis, drafting of the manuscript. D.T., F.E.: Study supervision. D.T.: Obtained funding

## Additional information

**Competing interests:** The authors declare no competing interests.

