## [Peer Review File · Nature Communications]

Reviewer #1 (Remarks to the Author):

Very nice work on the role of H2AZ in intestine epithelium homeostasis. Straightforward and coherent work with clear results. However, the mechanism by which H2AZ may repress gene expression is not clear and actually indirectly speculated from observations gathered in the absence of H2AZ. This should be supported biochemically (e.g. binding of CDX2 to recombinant nucleosomes with or without H2AZ).

That being said, as you may expect, there are a few issues that should be addressed:

- 1- line 106, "inducing a negative control" could be rephrase to "repressing transcription" in order to be clearer.
- 2- lines 138-141, to definitely show that H2AZ and p400 are required for proliferation and that the phenotype is not due to off-target effects, the authors should reconstitute their expression using siRNA-resistant H2AZ and p400 cDNAs or at least validate that H2AZ and p400 are reduced at the protein level.
- 3- line 150, H2AZ siRNA induces senescence and permanent cell cycle withdrawal in BC. Is it the case in intestinal epithelial cells?
- 4- as stated by the authors, (lines 168-170 "inducing positive or negative effects on gene expression depending on chromatin context as well as its post-translational modifications"), modifications of H2AZ rather than the presence of H2AZ at promoters regulate transcription, just like modifications on canonical H3. A potential molecular mechanism to explore here would be to show that H2AZKac or H2AZKme regulate intestinal differentiation genes.
- 5- line 189, CDX2 protein levels are not convincingly higher...
- 6- line 196, How come SI and LPH (2 CDX2 targets) are not induced upon siH2AZ when CDX2 expression is induced by DOX (Figure 2B)?
- 7- line 205, lack of molecular mechanism.... How does H2AZ regulates gene expression? Silencing of H2AZ enhances SI and LPH expression, indirectly suggesting that H2AZ represses their expression, but how? Acetylation, methylation, SUMOylation of H2AZ?
- 8- lines 234-236, Although "significant" the difference remains small compared to siH2AZ in Caco-2/15 cells (mRNA and protein levels Figure 2A)...
This is a bit worrisome especially given that the expression of Lph is unaffected in vivo while expression is induced in Caco cells...
- 9- line 238, not a knockdown, but a knockout....
- 10- line 243, not a knockdown, but a knockout...
- 11- line 266, the CDX2 knockdown is really inefficient at the mRNA level. This should be validated at the protein level before making any further conclusions.
- 12- line 269, CDX2-knockdown is NOT convincing, so I'm very uncomfortable with the CDX2-dependent conclusion.
- 13- line 281, specify that it's a decrease of H2AZ.
- 14- line 282, regarding the "local decrease", it is a MILD decrease. Also, how did the author calculate a p-value of 0.026 with the HUGE error bars on the MUCDHL promoter graph...?
- 15- line 286, the conclusion is a bit exaggerated! The results suggest that reduced H2AZ at promoters may favor CDX2 occupancy...
- 16- line 309, likewise, the conclusion overstates what the results show. Thus, these results prove suggest (at most...) that the reduction of H2AZ....

Hopefully these comments and suggestions will be constructive and help secure the publication of your article in Nat Comm or another good journal.

Best,
O.

Reviewer #2 (Remarks to the Author):

Little is known about functions of histone variants in regulation of adult tissue homeostasis. In this work, Rispal et al., use a combination of mouse genetics, in vitro cell culture assays and genomics to study functions of H2AZ during maintenance and differentiation of adult intestinal stem cells. The authors firstly generate the data from in vitro cell culture showing that H2AZ is required for

proliferation of normal and neoplastic cells. They observe an increase in the expression of some differentiation markers upon loss of H2Az. They show that a decrease in H2Az levels promotes activation of Cdx2 mediated transcription. Furthermore, overexpression of H2Az inhibits an activation of some genes upon loss of b-catenin. From these studies, the authors conclude that H2Az is a mediator of Wnt dependent repression of Cdx2 target genes.

Wnt signaling is always associated with an activation of gene expression. How and whether it can repress the other set of genes is not known. It is why the manuscript is interesting. Yet, the link between b-catenin and H2Az is very weak. The current study does not truly provide the mechanism. The loss of H2Az upon b-catenin depletion could be indirect. Furthermore, in vivo H2Az seems to be dispensable for ISC maintenance. Here, however, more thorough study is required to support the conclusions.

Major points:

1. The authors show that knocking down either p400 or H2Az using siRNA leads to reduction of the cell colonies (Figure 1). They suggest a link between loss of p400 and H2Az incorporation. Yet, the data supporting this claim are not provided. First, reduced cell numbers could be due to apoptosis. The authors should provide data describing the rate of cell proliferation, such as EdU/ BrdU incorporation or any other alternative marker. Moreover, they should check for apoptosis using either activated Casp3 antibody or TUNEL assay.

Second, immunostaining using H2Az antibody on the tested cell lines is necessary to confirm the effects of p400 loss and its link to H2Az incorporation.

2. Figure 1B is confusing. They authors test for cell proliferation and use differentiated cells that should not proliferate. They see some difference. I am not sure how statistically significant, though. Probably, the induction of the differentiation does not work 100% or it is an evidence of apoptosis. I would not include this experiment in the manuscript.

3. Figure 2. Knock-down of H2Az leads to activation of Cdx2, Sis and LPH expression in both Caco and HIEC. In Figure 5B, the authors provide ChIP data on the presence of H2Az at the promoters of LPH and MUCDHL but not of SI gene. In the mouse small intestine, neither Sis nor Lct contain H2Az at their promoters (Kazakevych et al., 2017, Figure 5). This is consistent with low enrichment of H2Az at LPH promoter compared to MUCDHL. Could authors test for the effect of H2Az loss on the expression of its targets, including Klf4, Arhgef2, and Ldha?

4. Lgr5-EGFP-Cre-ERT2 mouse model displays variegated expression of EGFP-CRE-ERT2 in the small intestinal epithelium. Around 20% of crypts are labelled only. H2Az+ cells could be progenies of ISCs in which recombination did not take place (mosaic cells in Figure 3A). Therefore, to make conclusions about functions of H2Az in vivo, careful examination for recombination events is required. First, double labelling for Ki67 and H2Az is required. The same is true for Lyz1 and goblet cells. Next, what will happen after 2 months post tamoxifen feeding? Will H2Az-negative crypts, ISCs be at the same numbers or they will be replaced by wt stem cells?

5. Could the authors comment on the nature of H2Az+ cells? Is it a specific cell type?

6. The authors selected B2m gene for normalization of their qPCR results. To date, it is not known whether B2m is affected by loss of H2Az. I suggest using "classical" genes for normalization of their results as Tbp or EpCAM. The figure 3 and 4a-d should be combined for easier reading and clarity.

7. The figure 4E and 5 should be combined for clarity. Here, again I suggest testing an enrichment for H2Az and Cdx2 at the promoters of Klf4, Arhgef2, and Ldha genes.

8. The effect of b-catenin depletion on the levels of H2Az might be indirect. B-catenin regulates cell proliferation. Therefore, it might be that a decrease in H2Az levels reflects cell cycle exit and differentiation. Here, to provide an evidence that H2Az is a direct target of b-catenin/Tcf712, the authors should make Tcf712 ChIP (or use publically available Tcf712 data). Next, they should delete the regulatory element containing Tcf712 site using CRISPR/ Cas9 in Caco cells.

Minor points:

9. ISCs divide symmetrically (lane 75, Introduction).

10. Please, change repression to activation in "Taken together, our data show that the repression of H2A.Z expression is one of the mechanisms by which the Wnt signaling pathway controls the progenitor maintenance and the differentiation of the intestinal epithelium." (lanes 316-318, Results)

11. I think the PCR analysis is misleading (Figure Suppl 3). Certainly, the larger non-recombined allele will have disadvantage for amplification compared to the shorter recombined bind. I think the authors should comment about it in the figure legend.

12. The description of the Lgr5 mouse is too long. The model is well known and was generated by the other group.

13. Overall, the text should be improved.

14. Figures should be better labelled. They should be put in order according to the text. It is very difficult to follow description of the results jumping between Figure 2 and 5. qPCR results in Figures 6-7 should be labelled with a,b,c,d.

We would like to thank the reviewers for their very helpful comments which undoubtedly improved our manuscript. Please find below a point-by-point answer to reviewers' comments.

Reviewer #1 (Remarks to the Author):

Very nice work on the role of H2AZ in intestine epithelium homeostasis. Straightforward and coherent work with clear results. However, the mechanism by which H2AZ may repress gene expression is not clear and actually indirectly speculated from observations gathered in the absence of H2AZ. This should be supported biochemically (e.g. binding of CDX2 to recombinant nucleosomes with or without H2AZ).

We would like to thank the reviewer for his very positive comments on our manuscript.

We agree with the reviewer that the mechanism involved in the H2A.Z-dependent repression of gene expression merited to be investigated further. We performed the requested experiment and found that both H2A- and H2A.Z-containing nucleosomes block binding of CDX2 to its target site. This experiment is now included in the revised manuscript as SupplInfo 13. Indeed, these data show that it is not by merely replacing canonical H2A that H2A.Z blocks CDX2 binding. We can thus envision three non exclusive possibilities:

- In the first one, the depletion of H2A.Z upon siRNA or upon differentiation leads to the decrease of local nucleosome occupancy, resulting in higher binding of CDX2 to its target sites. We thus performed H3 ChIP to monitor local nucleosome occupancy and found that H2A.Z depletion induces a significant reduction in H3 enrichment around the TSS of differentiation markers genes (SupplInfo 14). Thus, the eviction of the nucleosome in the CDX2-targetted regions of promoters can be, at least in part, involved in the induction of these genes.

- The second hypothesis is that post-translational modifications of H2A.Z blocks CDX2 recruitment, for example through the recruitment of a negative regulator of CDX binding. As stated by this reviewer, H2A.Z can be acetylated, sumoylated, and perhaps methylated, and these modifications in turn could be recognized by specific "readers" which could inhibits CDX2 binding (for example through an enzymatic modification of CDX2). We performed ChIP experiments using anti-acetylated H2A.Z antibody and we were not able to correlate the presence of this mark with a specific transcriptional response of differentiation-linked genes (see Data for reviewers 1). However, other post-translational modifications of H2A.Z could be involved.

- Finally, the third hypothesis is that in the presence of siRNA or upon differentiation, H2A.Z is replaced by H2A, which could modified in such a way that this modification would favor CDX2 recruitment, in a mechanism similar to what is described above.

Note however that investigating these latter mechanisms is beyond the scope of our manuscript: despite decades of strong efforts in the field, how histones and their post-translational modifications regulate transcription is still not fully understood.

We have included a full paragraph in the discussion section of the manuscript extensively discussing this very important point.

That being said, as you may expect, there are a few issues that should be addressed: 1- line 106, "inducing a negative control" could be rephrase to "repressing transcription" in order to be clearer.

The text has been modified, as suggested.

2- lines 138-141, to definitely show that H2AZ and p400 are required for proliferation and that the phenotype is not due to off-target effects, the authors should reconstitute their expression using siRNA-resistant H2AZ and p400 cDNAs or at least validate that H2AZ and p400 are reduced at the protein level.

Data concerning p400 have been removed from the manuscript (see below). Concerning H2A.Z, we performed the complementation experiment, which is now shown in SupplInfo 2. We have also included western-blot showing the efficiency of the siRNA-depletion of H2A.Z (SupplInfo 3A). The text has been changed accordingly.

3- line 150, H2AZ siRNA induces senescence and permanent cell cycle withdrawal in BC. Is it the case in intestinal epithelial cells ?

We thank the reviewer for this interesting question. To address it, we performed RT-qPCR and western blots to analyze the expression of senescence inducers or markers in siRNA-depleted Caco-2/15 and HIEC cells and these data are now included in the revised manuscript as SupplInfo 3. In our conditions, we did not observe any increase in expression of such markers, indicating that there is no massive induction of senescence which could explain the proliferation arrest of the cell population upon H2A.Z depletion. We also found that there is no detectable induction of apoptosis [SupplInfo 3], but we observed an accumulation of cells at the G1 phase of the cell cycle [SupplInfo 4], in agreement with a delay in cell cycle progression. These data are included in the revised manuscript as "Supplementary informations" and the text has been changed accordingly.

4- as stated by the authors, (lines 168-170 "inducing positive or negative effects on gene expression depending on chromatin context as well as its post-translational modifications"), modifications of H2AZ rather than the presence of H2AZ at promoters regulate transcription, just like modifications on canonical H3. A potential molecular mechanism to explore here would be to show that H2AZKac or H2AZKme regulate intestinal differentiation genes.

We agree with the reviewer that H2A.Z post-translational modifications could play an important role in its transcriptional functions. We performed ChIP experiments to monitor involvement of H2A.Z acetylation in transcription, and we did not find any correlation between changes in histone acetylation levels and transcription, neither during *in vitro* Caco-2-15 cells differentiation nor upon Tip60 knock-down (see Data for reviewers 1A and 1B). These data do not allow us to conclude on the involvement of H2A.Z acetylation in the transcriptional regulatory mechanism of the Caco-2/15 differentiation. But we cannot exclude any role of post-translational modifications in H2A.Z functions and, as stated above, we discuss the possible function of such modifications in the revised manuscript.

5- line 189, CDX2 protein levels are not convincingly higher...

We agree with the reviewer that the change in the expression of CDX2 mRNA is weak. We tempered the text on this point. However, we now show that it can also be observed by western blotting [Fig 2A and SupplInfo 6]. The text of the manuscript has been changed accordingly.

6- line 196, How come SI and LPH (2 CDX2 targets) are not induced upon siH2AZ when CDX2 expression is induced by DOX (Figure 2B)?

Our data show that H2A.Z induces a strong CDX2-dependent expression of SI and LPH without significantly increasing CDX2 itself, but probably regulating its binding to target sequences. We believe that, when Cdx2 is strongly overexpressed in the presence of Dox, CDX2-dependent activation of these target genes is maximal and cannot be further increased by H2A.Z depletion. This interpretation is now stated in the revised manuscript.

7- line 205, lack of molecular mechanism.... How does H2AZ regulates gene expression? Silencing of H2AZ enhances SI and LPH expression, indirectly suggesting that H2AZ represses their expression, but how? Acetylation, methylation, SUMOylation of H2AZ?

See our answer above.

8- lines 234-236, Although "significant" the difference remains small compared to siH2AZ in Caco-2/15 cells (mRNA and protein levels Figure 2A)...

This is a bit worrisome especially given that the expression of Lph is unaffected in vivo while expression is induced in Caco cells...

We agree with the referee about the differences on gene expression upon H2a.z reduction in mice are small. However, it has to be noted that, as mentioned by the reviewer 2 (see point 4 of his comments), the *"Lgr5-EGFP-Cre-ERT2 mouse model displays variegated expression of EGFP-CRE-ERT2 in the small intestinal epithelium. Around 20% of crypts are labelled only. H2Az+ cells could be progenies of ISCs in which recombination did not take place (mosaic cells in Figure 3A)."*

To document this point, we analysed the decrease of H2A.Z expression itself. Although we detected a strong decrease in the number of H2A.Z positive cells 10 days following Tamoxifen addition, this decrease is much weaker 1 month following Tamoxifen addition and completely absent after two months, indicating that wild type not recombined cells rapidly recolonized the intestine. This is even more striking when we analysed H2A.Z mRNA levels, for which we do not even observe a significant reduction, indicating that variations from mice to mice overcame the partial decrease due to incomplete recombination. Thus, incomplete recombination due to the variegated expression of the CRE recombinase probably explains the weakness of the transcriptional effects we observed upon Tamoxifen addition.

These data are included in the revised manuscript [SupplInfo 9] and the text has been changed accordingly.

9- line 238, not a knockdown, but a knockout....

The text has been modified, as suggested.

10- line 243, not a knockdown, but a knockout...

The text has been modified, as suggested.

11- line 266, the CDX2 knockdown is really inefficient at the mRNA level. This should be validated at the protein level before making any further conclusions.

As suggested, the CDX2 knockdown has been validated by western-blot in the revised manuscript (see SuppInfo 6). The text of the manuscript has been changed accordingly.

12- line 269, CDX2-knockdown is NOT convincing, so I'm very uncomfortable with the CDX2-dependent conclusion.

Accurate measurements of CDX2 mRNA by RT-qPCR were performed on 3 independent experiments and indicate a consistent 50% decrease, which is also observed at the protein level in the revised manuscript (see above). In addition, we observed the expected decrease of CDX2-target genes. We thus believe that valid conclusions can be drawn from CDX2 knockdown experiments.

13- line 281, specify that it's a decrease of H2AZ.

The text has been modified, as suggested.

14- line 282, regarding the "local decrease", it is a MILD decrease. Also, how did the author calculate a p-value of 0.026 with the HUGE error bars on the MUCDHL promoter graph...?

The reviewer is right in pointing out that the decrease is not very strong. We changed the text to tune down our conclusion. This mild decrease is probably due to the fact that histones are very stable when incorporated in chromatin. However, the data are very robust: Indeed, the huge error bars come to the fact that the % of input, which is the unprocessed result of ChIP experiments, varies from one experiment to the other. However, a similar effect of H2A.Z siRNA was observed in the three entirely independent experiments. This is shown by the low p value, which was calculated using a paired test, which allows comparing the effect of H2A.Z siRNA to control siRNA in individual experiments.

An alternate representation would be to standardized to 1 the percentage of input in the control sample in each experiment, and then showing the mean result for H2A.Z siRNA. In such a representation, the error bar for the H2A.Z siRNA point would be very small.

However, we decided to keep unprocessed data (% of input), which include more information than processed data (such as the % of input in classical experiments). We clearly indicate in the legend figure that the error bars came from variations in ChIP efficiency, but that the effect of H2A.Z siRNA was highly similar in all experiments.

15- line 286, the conclusion is a bit exaggerated! The results suggest that reduced H2AZ at promoters may favor CDX2 occupancy...

The text has been toned down, as suggested.

16- line 309, likewise, the conclusion overstates what the results show. Thus, these results prove suggest (at most...) that the reduction of H2AZ....

The text has been toned down, as suggested.

Hopefully these comments and suggestions will be constructive and help secure the publication of your article in Nat Comm or another good journal.

Reviewer #2 (Remarks to the Author) :

Little is known about functions of histone variants in regulation of adult tissue homeostasis. In this work, Rispal et al., use a combination of mouse genetics, in vitro cell culture assays and genomics to study functions of H2Az during maintenance and differentiation of adult intestinal stem cells. The authors firstly generate the data from in vitro cell culture showing that H2Az is required for proliferation of normal and neoplastic cells. They observe an increase in the expression of some differentiation markers upon loss of H2Az. They show that a decrease in H2Az levels promotes activation of Cdx2 mediated transcription. Furthermore, overexpression of H2Az inhibits an activation of some genes upon loss of b-catenin. From these studies, the authors conclude that H2Az is a mediator of Wnt dependent repression of Cdx2 target genes.

Wnt signaling is always associated with an activation of gene expression. How and whether it can repress the other set of genes is not known. It is why the manuscript is interesting. Yet, the link between b-catenin and H2Az is very weak. The current study does not truly provide the mechanism. The loss of H2Az upon β -catenin depletion could be indirect. Furthermore, in vivo H2Az seems to be dispensable for ISC maintenance. Here, however, more thorough study is required to support the conclusions.

We first would like to thank the reviewer for his enthusiastic feeling and his helpful analysis of our manuscript.

We agree with the reviewer that the mechanistic link between H2A.Z and Wnt pathway was weak in the first version of the manuscript. Thus, we now add experiments which support and reinforce our conclusions, as described below and discussed in the manuscript.

Major points:

1. The authors show that knocking down either p400 or H2Az using siRNA leads to reduction of the cell colonies (Figure 1). They suggest a link between loss of p400 and H2Az incorporation. Yet, the data supporting this claim are not provided. First, reduced cell numbers could be due to apoptosis. The authors should provide data describing the rate of cell proliferation, such as EdU/ BrdU incorporation or any other alternative marker. Moreover, they should check for apoptosis using either activated Casp3 antibody or TUNEL assay.

Second, immunostaining using H2Az antibody on the tested cell lines is necessary to confirm the effects of p400 loss and its link to H2Az incorporation.

We agree with the reviewer that the decrease we observed in proliferation of our models could be due to several anti-proliferative processes such as cell cycle arrest, apoptosis, senescence, ... Thus, as suggested, we performed an EdU incorporation experiment in Caco-2 cells after siRNA-mediated knockdown of Tip60, p400 or H2A.Z. Our results showed that p400 knockdown induces an accumulation of cells in G2 phase (see SupplInfo 4), whereas H2A.Z siRNA seems to block the cell cycle in G0/G1 phase.

The G2 accumulation of p400 depleted cells are consistent with our previous finding that p400 depletion induced a cell cycle arrest due to the activation of the DNA damage response pathway (Mattera et al., 2010): since Caco2 cells have a p53^{mut} genotype, activation of the DDR pathway leads to an arrest in the G2 phase and not in G0/G1.

The G0/G1 accumulation observed upon H2A.Z depletion is in agreement, with the G0 exit of the cell cycle and the increase of the differentiation marker expression.

Because these data seem to demonstrate that the mechanism involved upon p400 depletion does not seem to be dependent on H2A.Z, we removed p400 data from Figure 1. This does not change the main conclusions from our study, since all the other experiments were focused on H2A.Z.

We also assayed induction of apoptosis as requested by this reviewer and senescence following the other reviewer request. We did not observe any induction of apoptosis neither of senescence upon H2A.Z depletion. These data are included in the revised manuscript in SupplInfo 3 and the text has been changed accordingly.

We would like to thank the reviewer for having pointed out this weakness in our manuscript.

2. Figure 1B is confusing. They authors test for cell proliferation and use differentiated cells that should not proliferate. They see some difference. I am not sure how statistically significant, though. Probably, the induction of the differentiation does not work 100% or it is an evidence of apoptosis. I would not include this experiment in the manuscript.

We thank the reviewer for this pertinent argument. We agree that this panel did not reinforce the message of the figure and we suppressed it, as suggested.

3. Figure 2. Knock-down of H2Az leads to activation of Cdx2, Sis and LPH expression in both Caco and HIEC. In Figure 5B, the authors provide ChIP data on the presence of H2Az at the promoters of LPH and MUCDHL but not of SI gene. In the mouse small intestine, neither Sis not Lct contain H2Az at their promoters (Kazakevych et al., 2017, Figure 5). This is consistent with low enrichment of H2Az at LPH promoter compared to MUCDHL. Could authors test for the effect of H2Az loss on the expression of its targets, including Klf4, Arhgef2, and Ldha?

As suggested, we tested the expression of other previously described genes bound by H2A.Z. In Caco-2/15 cells, we observed that, among suggested targets, KLF4 is indeed induced upon H2A.Z knockdown [SupplInfo 7A], as well as upon treatment with β -catenin siRNA [SupplInfo 7C]. The other two genes were unchanged, which is not really surprising given that the ability to respond to H2A.Z depletion is not always correlated to the binding of H2A.Z to gene promoters (Coleman-Derr, 2012). Strikingly, KLF4 is known to be regulated by Cdx2 (Faber, 2013), which reinforce the link between activation upon H2A.Z depletion and regulation by CDX2. These data are now included in the manuscript as SupplInfo 7 and the text has been changed accordingly.

4. Lgr5-EGFP-Cre-ERT2 mouse model displays variegated expression of EGFP-CRE-ERT2 in the small intestinal epithelium. Around 20% of crypts are labelled only. H2Az+ cells could be progenies of ISCs in which recombination did not take place (mosaic cells in Figure 3A). Therefore, to make conclusions about functions of H2Az in vivo, careful examination for recombination events is required. First, double labelling for Ki67 and H2Az is required. The same is true for Lyz1 and goblet cells. Next, what will happen after 2 months post tamoxifen feeding? Will H2Az-negative crypts, ISCs be at the same numbers or they will be replaced by wt stem cells?

We agree with the reviewer that the absence of effect on Ki67 and Lyz1 had to be correlated with H2A.Z staining. As requested, the labelling on serial sections of H2A.Z and these markers has been done and panels are now included in the Figure 3C.

As also suggested by the reviewer, we analyzed the long-term maintenance of this expression pattern: we found that 1 month following Tamoxifen addition, crypt cells are again positive for H2A.Z, and 2 months following Tamoxifen addition, we no longer detect any H2A.Z negative cells in the intestine [SuppInfo 9]. These data indicate that wild type stem cells probably replace knock-out cells, indicating that, despite we do not detect any phenotype, H2A.Z knock-out stem cells suffer from proliferation defects, in agreement with the interpretation that H2A.Z is required for stem cells maintenance. These data are included in the revised manuscript and the text has been changed accordingly.

5. Could the authors comment on the nature of H2Az+ cells? Is it a specific cell type?

The apparently higher expression of H2A.Z in these cells that could be observed in images of the original manuscript was misleading, since it was not consistently observed in knock-out mice. We made this point clear in the revised manuscript. Since we did not observe any aberrant or ectopic expression in these cells (such as Ki67 for example), these cells probably comes from incomplete recombination of H2A.Z-encoding gene. This explanation is now mentioned in the revised manuscript.

6. The authors selected B2m gene for normalization of their qPCR results. To date, it is not known whether B2m is affected by loss of H2Az. I suggest using “classical” genes for normalization of their results as Tbp or EpCAM.

We agree with the reviewer that standardization of such results is a fundamental point. This is why we already used two genes for normalizing our data in the original manuscript: β 2m and RPLP0. But as suggested, we also analyze the relative variation of TBP and GAPDH mRNA expression towards β 2m, in both mice and Caco-2/15 cells, and in control or H2A.Z depleted conditions (see Data for reviewers 2). We did not observed any significant variation of these genes normalized to β 2m and RPLP0 whatever the model or the condition.

Thus, we conclude on the validity of our results presented in the initial version of the manuscript.

The figure 3 and 4a-d should be combined for easier reading and clarity.

We agree with the reviewer that panels regarding the same gene at both mRNA and protein expression levels could be re-arranged in the figures, and we tried to follow this reviewer suggestion. However, we found it difficult to perform since we analyzed a much larger number of markers by RT-qPCR than at protein level. We found that the clarity of the results presentation was even worse and thus decided to conserve the original presentation.

7. The figure 4E and 5 should be combined for clarity.

We believe that combining data presenting the expression of intestine specific transcription factors at mRNA levels in mice intestines upon KO for H2A.Z (Figure 4E) with the ones showing the results of RT-qPCR upon H2A.Z and/or CDX2 siRNA transfection and CHIP for H2A.Z and CDX2 upon H2A.Z

depletion all obtained in Caco-2/15 cells (Figure 5) may cause confusion when reading the manuscript. Thus, we decided to conserve their original presentation.

Here, again I suggest testing an enrichment for H2Az and Cdx2 at the promoters of Klf4, Arhgef2, and Ldha genes.

As suggested, we tested the enrichment in CDX2 of the KLF4 promoter in Caco-2/15 cells and we observed that H2A.Z knockdown induces a drastic increase of the binding of CDX2 transcription factor [SuppInfo 7B]. Thus, it strongly suggested that we highlighted a general mechanism which can be extrapolated to other genes than differentiation markers. Note that when assaying H2A.Z occupancy, we could not detect any decrease perhaps because variations between CHIP efficiency or in qPCR measurements overcame the variations in H2A.Z occupancy. Alternatively, this lack of H2A.Z binding decrease could reflect an indirect regulation of KLF4 by H2A.Z.

These data are now included in the manuscript in SuppInfo 7 and the text of the manuscript was changed accordingly.

8. The effect of b-catenin depletion on the levels of H2Az might be indirect. B-catenin regulates cell proliferation. Therefore, it might be that a decrease in H2Az levels reflects cell cycle exit and differentiation. Here, to provide an evidence that H2Az is a direct target of b-catenin/Tcf7l2, the authors should make Tcf7l2 CHIP (or use publically available Tcf7l2 data). Next, they should delete the regulatory element containing Tcf7l2 site using CRISPR/ Cas9 in Caco cells.

We agree with the reviewer that the effects of β -catenin depletion could be indirect.

We thus analyzed published TCF7L2 ChIP-seq data (Frietze et al. 2012) and observed the presence of a strong signal around the promoter of the H2A.Z-encoding gene in HCT116 cells, which, like Caco2 cells, are derived from colon carcinoma [Figure 7A].

We thus performed CHIP experiments in both HCT116 and Caco-2/15 cells and we observed that TCF7L2 binds to H2A.Z promoter [Figure 7C]. This provides evidence that H2A.Z is a direct target of TCF7L2 and could be directly regulated by the Wnt pathway.

To test this possibility, we established a cell line derived from HCT116 cells and harbouring CRISPR/Cas9-mediated deletion of the upstream major Tcf7l2 binding site (identified from the published CHIP-Seq experiments) within H2A.Z promoter. We used HCT116 cells rather than Caco2/15 cells given that they are easier to genetically manipulate. Moreover, the H2A.Z gene is also regulated by the Wnt pathway in these cells, since we observed that, as we described in Caco-2 cells in the original manuscript, depletion of β -catenin results in a decrease of H2A.Z expression [Figure 7B].

We then confirmed that the binding of TCF7L2 is strongly affected in the promoter-deleted clone [Figure 7D] and analyzed the effect of β -catenin depletion on the expression of H2A.Z in control cells or cells harboring TCF7L2 binding site deletion in H2A.Z promoter [Figure 7E]. We observed that, in cells harboring the deletion, depletion of β -catenin led to a weaker decrease of H2A.Z expression than control cells, whereas the effect of β -catenin depletion on another Wnt-target genes was similar. The effect is only partial probably because there is still residual binding of TCF to another site downstream of H2A.Z promoter and observed in CHIP Seq data and that we could not delete without deleting the H2A.Z promoter and coding region.

Altogether, these data showed that H2A.Z is a direct target of and is induced by the Wnt pathway. They are included in the manuscript and the text has been changed accordingly. Again, we are grateful to this reviewer for very pertinent comment which helped us improving the manuscript.

Minor points :

9. ISCs divide symmetrically (lane 75, Introduction).

The text has been modified, as suggested.

10. Please, change repression to activation in “Taken together, our data show that the repression of H2A.Z expression is one of the mechanisms by which the Wnt signaling pathway controls the progenitor maintenance and the differentiation of the intestinal epithelium.” (lanes 316-318, Results)

The text has been modified, as suggested.

11. I think the PCR analysis is misleading (Figure Suppl 3). Certainly, the larger non-recombined allele will have disadvantage for amplification compared to the shorter recombined bind. I think the authors should comment about it in the figure legend.

The text of the figure legend has been modified, as suggested.

12. The description of the Lgr5 mouse is too long. The model is well known and was generated by the other group.

The text has been modified, as suggested.

13. Overall, the text should be improved.

The text has been modified, as suggested.

14. Figures should be better labelled. They should be put in order according to the text. It is very difficult to follow description of the results jumping between Figure 2 and 5. qPCR results in Figures 6-7 should be labelled with a,b,c,d.

Figures labelling has been modified, as suggested.

We agree with the reviewer that it would be better to have data nearly referred in text in the same figure, but for the logical organization of the manuscript, mechanistic aspects of the transcriptional regulation by H2A.Z have to be presented after the *in vivo* characterization of these effects. Thus, we decided to keep the same order in the figures presentation.

Data for reviewers 1A

Data for reviewers 1B

Data for reviewers 1 : Impact of H2A.Z acetylation on mRNA expression

A) RT-qPCR to analyze *p21* and *LPH* expression, and ChIP experiments for assaying total or acetylated H2A.Z on promoters of these genes, have been performed on mRNA and chromatin, respectively, of Caco-2/15 cells harvested at subconfluence or 7 days post-confluence stages. mRNA expression or H2A.Z enrichment on promoters of indicated genes were analyzed from 3 independent experiments. Results are presented as mean and standard deviation for mRNA analysis and a representative ChIP experiment is shown. B) Caco-2/15 cells were transfected using siRNA against Tip60 messenger and mRNA or chromatin were analyzed by qPCR and ChIP respectively, as in A.

Note that, during in vitro Caco-2-15 cells differentiation, there is no change of H2A.Z acetylation on the LPH promoter despite the activation of LPH expression. However, in the same time, H2A.Z enrichment decreases suggesting a hyper-acetylation of the residual chromatin-bound variant (Data for reviewers 1A). Moreover, we also tested the effect of the knock-down of Tip60, a HAT known to modify H2A.Z. Despite the fact that Tip60 depletion decreases H2A.Z acetylation, and not H2A.Z itself, at LPH promoter, it did not induce any drastic change in LPH expression. In addition, we observed that the siRNA-mediated depletion of Tip60 decreased both recruitment and acetylation of H2A.Z on p21 promoter in Caco-2/15 cells, suggesting that the presence of this histone variant is more decisive in transcriptional regulation than its modification, in this context. This is reinforced by the fact that H2A.Z acetylation appears unchanged on MUCDHL promoter, whereas the reduction of H2A.Z enrichment correlates with transcriptional effects.

Data for reviewers 2

Data for reviewers 2 : Analysis of TBP and P0 mRNA expression

mRNA expression of RPLP0 (P0) and/or TBP were analyzed in samples from mice and Caco-2/15 as described in the manuscript and results were presented as Figures 4 and 2A, respectively.

Reviewer #1 (Remarks to the Author):

Good job. I feel that the authors have addressed the issues that I have raised.
Best,
Olivier BINDA

Reviewer #2 (Remarks to the Author):

The authors successfully addressed my comments. The manuscript is significantly improved and merits publication in Nature Communications. The text would need editorial corrections.

Point-by-point response to reviewers

REVIEWERS' COMMENTS:

Reviewer #1 (Remarks to the Author):

Good job. I feel that the authors have addressed the issues that I have raised.

Best,

Olivier BINDA

- We thank the reviewer for his helpful comments on the original version of the manuscript that allowed us to significantly improve the quality of the paper.

Reviewer #2 (Remarks to the Author):

The authors successfully addressed my comments. The manuscript is significantly improved and merits publication in Nature Communications. The text would need editorial corrections.

- We greatly appreciated all pertinent comments of the reviewer on our original manuscript and we thank him for his help in the improvement of the quality of this publication.